# Free volume theory explains the unusual behavior of viscosity in a non-confluent tissue during morphogenesis

Rajsekhar Das[1], Sumit Sinha[2], Xin Li[1], TR Kirkpatrick[3], D Thirumalai[1,2]*

[1]Department of Chemistry, University of Texas at Austin, Austin, United States; [2]Department of Physics, University of Texas at Austin, Austin, United States; [3]Institute for Physical Science and Technology, University of Maryland, College Park, United States

**Abstract** A recent experiment on zebrafish blastoderm morphogenesis showed that the viscosity ($\eta$) of a non-confluent embryonic tissue grows sharply until a critical cell packing fraction ($\phi_S$). The increase in $\eta$ up to $\phi_S$ is similar to the behavior observed in several glass-forming materials, which suggests that the cell dynamics is sluggish or glass-like. Surprisingly, $\eta$ is a constant above $\phi_S$. To determine the mechanism of this unusual dependence of $\eta$ on $\phi$, we performed extensive simulations using an agent-based model of a dense non-confluent two-dimensional tissue. We show that polydispersity in the cell size, and the propensity of the cells to deform, results in the saturation of the available free area per cell beyond a critical packing fraction. Saturation in the free space not only explains the viscosity plateau above $\phi_S$ but also provides a relationship between equilibrium geometrical packing to the dramatic increase in the relaxation dynamics.

*For correspondence: dave.thirumalai@gmail.com

**Competing interest:** The authors declare that no competing interests exist.

## eLife assessment

This **fundamental** study substantially advances our physical understanding of the sharp increase and saturation of the viscosity of non-confluent tissues with increasing cell density. Through the analysis of a simplified model, this study provides **compelling** evidence that polydispersity in cell size and the softness of cells together can lead to this phenomenon. The work will be of general interest to biologists and biophysicists working on development.

## Introduction

There is great interest in characterizing the mechanical and dynamical properties of embryonic tissues because they regulate embryo development (*Kimmel et al., 1995*; *Keller et al., 2008*; *Petridou and Heisenberg, 2019*; *Hannezo and Heisenberg, 2019*; *Autorino and Petridou, 2022*). Measurements of bulk properties, such as viscosity and elastic modulus, and the dynamics of individual cells through imaging techniques, have been interpreted by adapting concepts developed to describe phase transitions (PTs), glass transition, and active matter (*Shaebani et al., 2020*; *Marchetti et al., 2013*; *Kirkpatrick and Thirumalai, 2015*; *Bär et al., 2020*).

Several experiments have shown that during embryo morphogenesis, material properties of the tissues change dramatically (*Morita et al., 2017*; *Mongera et al., 2018*; *Barriga et al., 2018*; *Petridou et al., 2019*; *Petridou et al., 2021*). Of relevance to our study is a remarkable finding that provided evidence that a PT occurs during zebrafish blastoderm morphogenesis, which was analyzed using rigidity percolation theory (*Petridou et al., 2021*; *Jacobs and Thorpe, 1995*; *Jacobs and Thorpe, 1996*; *Jacobs and Hendrickson, 1997*). The authors also estimated the viscosity ($\eta$) of the blastoderm

tissue using the micropipette aspiration technique (*Guevorkian et al., 2010*; *Petridou et al., 2019*). It was found that change in $\eta$ is correlated with cell connectivity ($\langle C \rangle$), rising sharply over a narrow range of $\langle C \rangle$. Surprisingly, a single geometrical quantity, the cell–cell contact topology controls both the rigidity PT and changes in $\eta$ in this non-confluent tissue, thus linking equilibrium and transport properties.

Here, we focus on two pertinent questions that arise from the experiments on zebrafish blastoderm. First, experiments (*Sinha and Thirumalai, 2021*) showed that $\eta$ increases as a function of the cell packing fraction ($\phi$) till $\phi \leq 0.87$. The dependence of $\eta$ on $\phi$ follows the well-known Vogel–Fulcher–Tammann (VFT) law (*Sinha and Thirumalai, 2021*), which predicts that $\eta$ grows monotonically with $\phi$. The VFT law, which is commonly used to analyze the viscosity of a class of glass-forming materials (*Angell, 1991*), is given by $\eta \sim \exp\left[\frac{1}{\phi_0/\phi - 1}\right]$, where $\phi_0$ is a constant. Surprisingly, for packing fractions, $\phi \geq \phi_S \approx 0.90$, $\eta$ deviates from the VFT law and is *independent* of $\phi$, which cannot be explained using conventional theories for glasses (*Berthier and Biroli, 2011*; *Kirkpatrick and Thirumalai, 2015*). Second, the experimental data (*Petridou et al., 2021*) was interpreted using equilibrium rigidity percolation theory (*Jacobs and Thorpe, 1995*; *Jacobs and Thorpe, 1996*; *Jacobs and Hendrickson, 1997*) for an embryonic tissue in which cells undergo random cell divisions. A priori it is unclear why equilibrium concepts should hold in zebrafish morphogenesis, which one would expect is controlled by non-equilibrium processes such as self-propulsion, growth, and cell division.

We show that the two conundrums (saturation of $\eta$ at high packing fractions and the use of equilibrium statistical mechanics in a growing system to explain PT) may be rationalized by (i) assuming that the interactions between the cells are soft, and (ii) the cell sizes are highly heterogeneous (polydisperse), which is the case in zebrafish blastoderm. Using an agent-based (particle) simulation model of a two-dimensional (2D) non-confluent tissue, we explore the consequences of varying $\phi$ (see 'Materials and methods' for the definition) of interacting self-propelled polydisperse soft cells, on $\eta$. The central results of our study are (i) the calculated effective viscosity $\bar{\eta}$ (for details see, Appendix 6, 'Dynamical changes in local packing fraction cause jammed cells to move'), for the polydisperse cell system, shows that for $\phi \leq \phi_S \approx 0.90$ the increase in viscosity follows the VFT law. Just as in experiments, $\eta$ is essentially independent of $\phi$ at high ($\geq \phi_S$) packing fractions. (ii) The unusual dependence of $\eta$ at $\phi \geq \phi_S$ is quantitatively explained using the notion of available free area fraction ($\phi_{\text{free}}$), which is the net void space that can be explored by the cells when they are jammed. At high densities, a given cell requires free space in order to move. The free area is created by movement of the neighboring cells into the available void space. One would intuitively expect that the $\phi_{\text{free}}$ should decrease with increasing packing fractions due to cell jamming, which should slow down the overall dynamics. Indeed, we find that $\phi_{\text{free}}$ decreases with increasing packing fraction ($\phi$) until $\phi_S$. The simulations show that when $\phi$ exceeds $\phi_S$, the free area $\phi_{\text{free}}$ saturates because the soft cells (characterized by 'soft deformable disks') can overlap with each other, resulting in the collective dynamics of cells becoming independent of $\phi$ for $\phi \geq \phi_S$. As a consequence, $\eta$ saturates at high $\phi$. (iii) Cells whose sizes are comparable to the available free area move almost like a particle in a liquid. The motility of small-sized cells facilitates adjacent cells to move through multi-cell rearrangement even in a highly jammed environment. The facilitation mechanism, invoked in glassy systems (*Biroli and Garrahan, 2013*), allows large cells to move with low mobility. A cascade of such facilitation processes enable all the cells to remain dynamic even above the onset packing fraction of the PT. (iv) We find that the relaxation time does not depend on the waiting time for measurements even in the regime where viscosity saturates. In other words, there is no evidence of aging even in the regime where viscosity saturates. Strikingly, the tissue exhibits ergodic (*Thirumalai et al., 1989*) behavior at all densities. The cell-based simulations, which reproduce the salient experimental features, may be described using equilibrium statistical mechanics, thus providing credence to the use of cell contact mechanics to describe both rigidity PT and dynamics in an evolving non-confluent tissue (*Petridou et al., 2021*).

## Results

### Experimental results

We first describe the experimental observations, which serve as the basis for carrying out the agent-based simulations. *Figure 1A* shows the bright-field images of distinct stages during zebrafish morphogenesis. A 2D section of zebrafish blastoderm (*Figure 1B*) shows that there is considerable dispersion

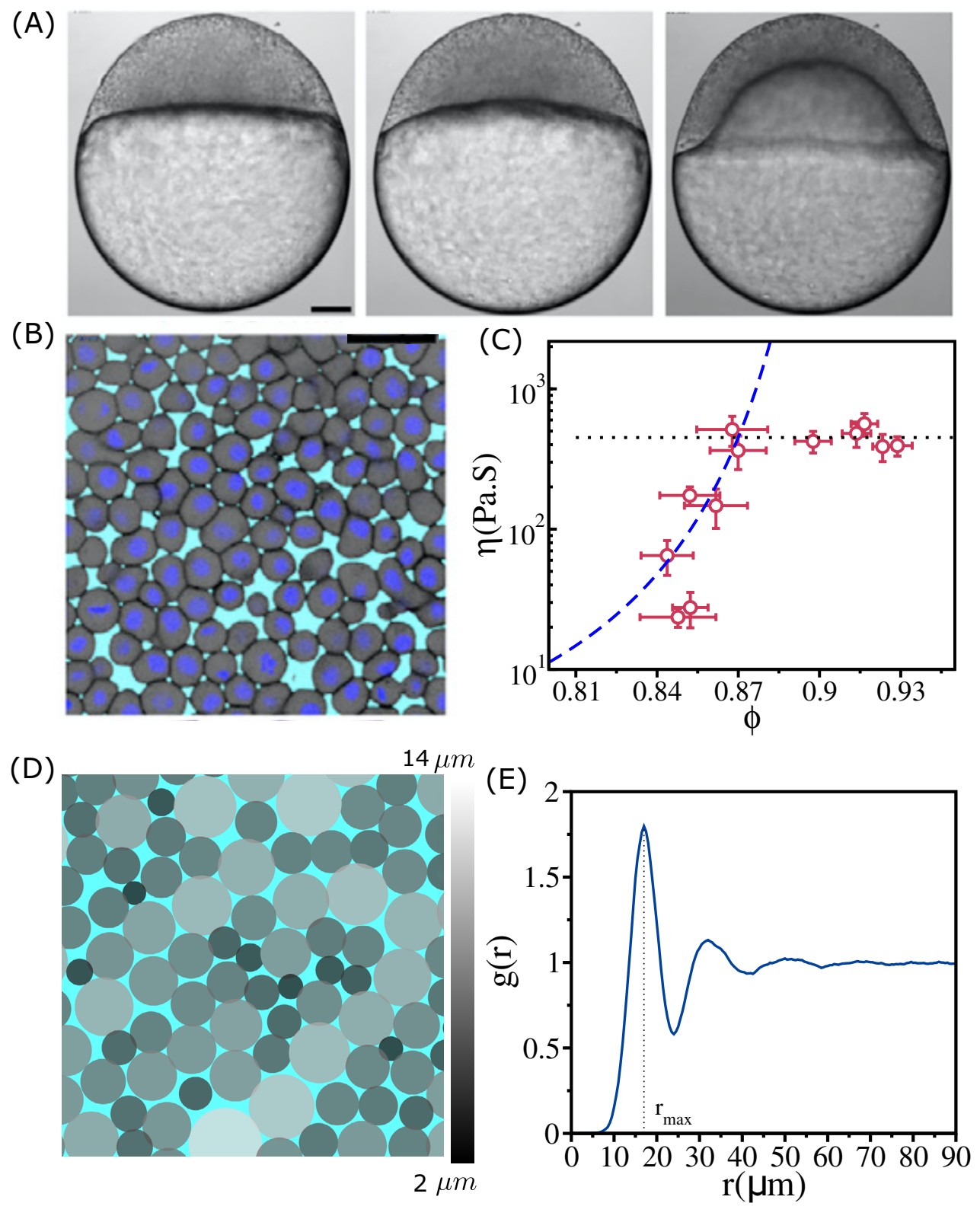

**Figure 1.** Structure and viscosity of non-confluent tissues. (**A**) Bright-field single-plane images of an exemplary embryo of zebrafish before ($t = -60$ min), at the onset ($t = 0$ min), and after blastoderm spreading ($t = 60$ min). (**B**) Snapshot of 2D confocal sections at the 1st–2nd deep-cell layer of the blastoderm at $t = 60$ min. (**A**) and (**B**) are taken from *Petridou et al., 2021*. (**C**) Viscosity $\eta$ of zebrafish blastoderm as a function of $\phi$ in a log-linear scale using the data from *Petridou et al., 2021*. The dashed line is the fit to Vogel–Fulcher–Tammann (VFT) equation. Note that $\eta$ does not change

*Figure 1 continued on next page*

*Figure 1 continued*

significantly beyond $\phi \geq 0.87$. (**D**) A typical snapshot taken from cell-based simulations for $\phi = 0.93$. Cells are colored according to their radii (in μm) (color bar shown on the right). (**E**) The pair correlation function, $g(r)$, as a function of $r$ for $\phi = 0.93$. The vertical dashed line is the position of the first peak ($r_{\max} = 17.0~\mu m$). The pair correlation function does not exhibit signs of long-range order. Scale bars in (**A**) is 100 μm and (**B**) is 50 μm.

in cell sizes. The statistical properties of the cell sizes are shown in *Appendix 1—figure 1D*. *Figure 1C* shows that $\eta$ increases sharply over a narrow $\phi$ range and saturates when $\phi$ exceeds $\phi_S \approx 0.90$.

To account for the results in *Figure 1C*, we first simulated a mono-disperse system in which all the cells have identical radius ($R = 8.5~\mu m$). Because the system crystallizes (*Appendix 1—figure 1A and B*), we concluded that the dynamics observed in experiments cannot be explained using this model. A 1:1 binary mixture of cells with different radii gives glass-like behavior for all $\phi$, with the relaxation time $\tau_\alpha$ as well as the effective viscosity $\bar{\eta}$ (defined in *Equation 1*) following the VFT behavior (see Appendix 2).

### Polydispersity and cell–cell interactions

In typical cell tissues, and zebrafish in particular, there is a great dispersion in the cell sizes, which vary in a single tissue by a factor of ~5–6 (*Petridou et al., 2021*; *Figure 1B*, *Appendix 1—figure 1D*). In addition, the elastic forces characterizing cell–cell interactions are soft, which implies that the cells can overlap, with $r_{ij} - (R_i + R_j) < 0$ when they are jammed (*Figure 1B, D*). Thus, both polydispersity (PD) and soft interactions between the cells must control the relaxation dynamics. To test this proposition, we simulated a highly polydisperse system (PDs) in which the cell sizes vary by a factor of ~8 (*Figure 1D* , *Appendix 1—figure 1E*).

A simulation snapshot (*Figure 1D*) for $\phi = 0.93$ shows that different sized cells are well mixed. In other words, the cells do not phase separate. The structure of the tissue can be described using the pair correlation function, $g(r) = \frac{1}{\rho} \left\langle \frac{1}{N} \sum_i^N \sum_{j\neq i}^N \delta\left(r - |\vec{r}_i - \vec{r}_j|\right) \right\rangle$, where $\rho = \frac{N}{L^2}$ is the number density, $\delta$ is the Dirac delta function, $\vec{r}_i$ is the position of the $i$th cell, and the angular bracket $\langle\rangle$ denotes an average over different ensembles. The $g(r)$ function (*Figure 1E*) has a peak around $r \sim 17\mu m$, which is approximately the average diameter of the cells. The absence of peaks in $g(r)$ beyond the second one suggests there is no long-range order. Thus, the polydisperse cell system exhibits liquid-like structure even at the high $\phi$.

### Effective shear viscosity($\bar{\eta}$) as a function of $\phi$

A fit of the experimental data for $\eta$ using the VFT (*Tammann and Hesse, 1926*; *Fulcher, 1925*) relation in the range $\phi \leq 0.87$ (*Figure 1C*) yields $\phi_0 \approx 0.95$ and $D \approx 0.51$ (*Sinha and Thirumalai, 2021*). The VFT equation for cells, which is related to the Doolittle equation (*White and Lipson, 2016*) for fluidity ($\frac{1}{\eta}$) that is based on free space available for motion in an amorphous system (*Doolittle and Doolittle, 1957*; *Cohen and Turnbull, 1959*), is $\eta = \eta_0 \exp\left[\frac{D}{\phi_0/\phi - 1}\right]$, where $D$ is the apparent activation energy. In order to compare with experiments, we calculated an effective shear viscosity ($\bar{\eta}$) for the polydisperse system using a Green–Kubo-type relation (*Hansen and McDonald, 2013*)

$$\bar{\eta} = \int_0^\infty dt \sum_{(\mu\nu)} \langle P_{\mu\nu}(t) P_{\mu\nu}(0)\rangle . \tag{1}$$

The stress tensor $P_{\mu\nu}(t)$ in the above equation is

$$P_{\mu\nu}(t) = \frac{1}{A}\left(\sum_{i=1}^N \sum_{j>i}^N \vec{r}_{ij,\mu} \vec{f}_{ij,\nu}\right), \tag{2}$$

where $\mu, \nu \in (x, y)$ are the Cartesian components of coordinates, $\vec{r}_{ij} = \vec{r}_i - \vec{r}_j$, $\vec{f}_{ij}$ is the force between $i$th and $j$th cells, and $A$ is the area of the simulation box. Note that $\bar{\eta}$ should be viewed as a proxy for shear viscosity because it does not contain the kinetic term and the factor $\frac{A}{k_B T}$ is not included in *Equation (1)* because temperature is not a relevant variable in the highly over-damped model for cells considered here.

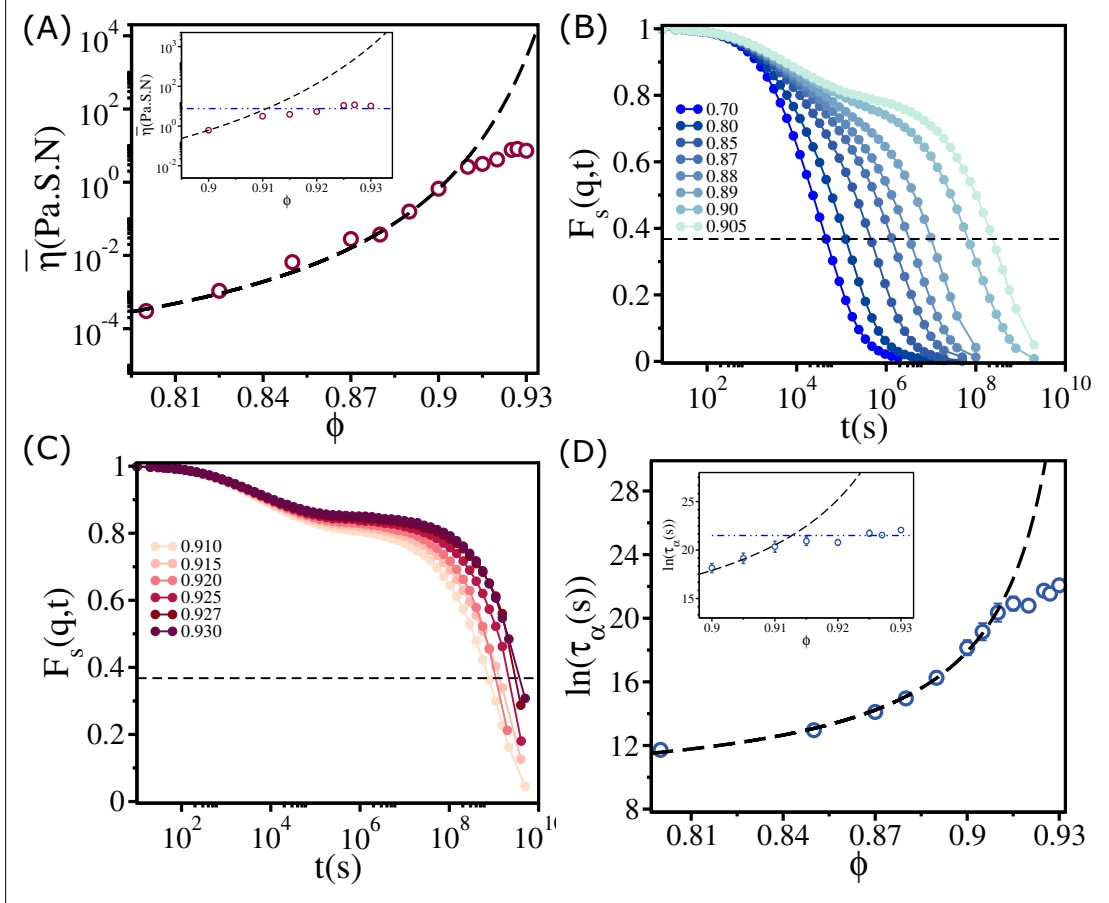

**Figure 2.** Saturation in viscosity and relaxation time. (**A**) Effective viscosity $\bar{\eta}$ as a function of $\phi$, with the solid line being the fit to Vogel–Fulcher–Tammann (VFT) equation. The inset shows $\bar{\eta}$ at high $\phi$. The dashed line in the inset is the expected behavior assuming that the VFT relation holds at all $\phi$. (**B**) The self-intermediate scattering function $F_s(q,t)$ as a function of $t$ for $0.70 \leq \phi \leq 0.905$. The dashed line corresponds to $F_s(q,t) = \frac{1}{e}$. (**C**) A similar plot for $\phi > 0.905$. (**D**) The logarithm of the relaxation time $\tau_\alpha(s)$ as a function of $\phi$. The VFT fit is given by the dashed line. The inset shows a zoomed-in view for $\phi \geq \phi_S$. The error bars in (**D**) are calculated using the standard deviation of $\tau_\alpha$ for 24 independent simulations.

Plot of $\bar{\eta}$ as a function of $\phi$ in *Figure 2A* shows qualitatively the same behavior as the estimate of viscosity (using dimensional arguments) made in experiments. Two features about *Figures 1C and 2A* are worth noting. (i) Both simulations and experiments show that up to $\phi \approx 0.90$, $\bar{\eta}(\phi)$ follows the VFT relation with $\phi_0 \sim 0.94$ and $D \sim 0.5$. More importantly, $\bar{\eta}$ is independent of $\phi$ when $\phi > 0.90$. (ii) The values of $\phi_0$ and $D$ obtained by fitting the experimental estimate of $\eta$ to the VFT equation and simulation results are almost identical. Moreover, the onset of the plateau packing fraction in simulations and experiments occurs at the same value ($\phi_S \sim 0.90$). The overall agreement with experiments is remarkable given that the model was not created to mimic the zebrafish tissue.

To provide additional insights into the dynamics, we calculated the isotropic self-intermediate scattering function, $F_s(q,t)$,

$$F_s(q,t) = \frac{1}{N} \left\langle \sum_{j=1}^{N} \exp[-i\vec{q} \cdot (\vec{r}_j(t) - \vec{r}_j(0))] \right\rangle, \tag{3}$$

where $\vec{q}$ is the wave vector, and $\vec{r}_j(t)$ is the position of a cell at time $t$. The degree of dynamic correlation between two cells can be inferred from the decay of $F_s(q,t)$. The angle bracket $\langle ... \rangle$ is an average over different time origins and different trajectories. We chose $q = \frac{2\pi}{r_{max}}$, where $r_{max}$ is the position of the first peak in $g(r)$ between all cells (see *Figure 1E*). The relaxation time $\tau_\alpha$ is calculated using $F_s(q, t = \tau_\alpha) = \frac{1}{e}$.

From *Figure 2B and C*, which show $F_s(q,t)$ as a function of $t$ for various $\phi$, it is clear that the dynamics become sluggish as $\phi$ increases. The relaxation profiles exhibit a two-step decay with a plateau in the intermediate time scales. The dynamics continues to slow down dramatically until $\phi \leq 0.90$. Surprisingly, the increase in the duration of the plateau in $F_s(q,t)$ ceases when $\phi$ exceeds $\approx 0.90$ (*Figure 2C*), a puzzling finding that is also reflected in the dependence of $\tau_\alpha$ on $\phi$ in *Figure 2D*. The relaxation time increases dramatically, following the VFT relation, till $\phi \approx 0.90$, and subsequently reaches a plateau (see the inset in *Figure 2D*).

If the VFT relation continued to hold for all $\phi$, as in glasses or in binary mixture of 2D cells (see Appendix 2), then the fit yields $\phi_0 \approx 0.95$ and $D \approx 0.50$. However, the simulations show that $\tau_\alpha$ is nearly a constant when $\phi$ exceeds 0.90. We should note that the behavior in *Figure 2D* differs from the dependence of $\tau_\alpha$ on $\phi$ for 2D monodisperse polymer rings, used as a model for soft colloids.

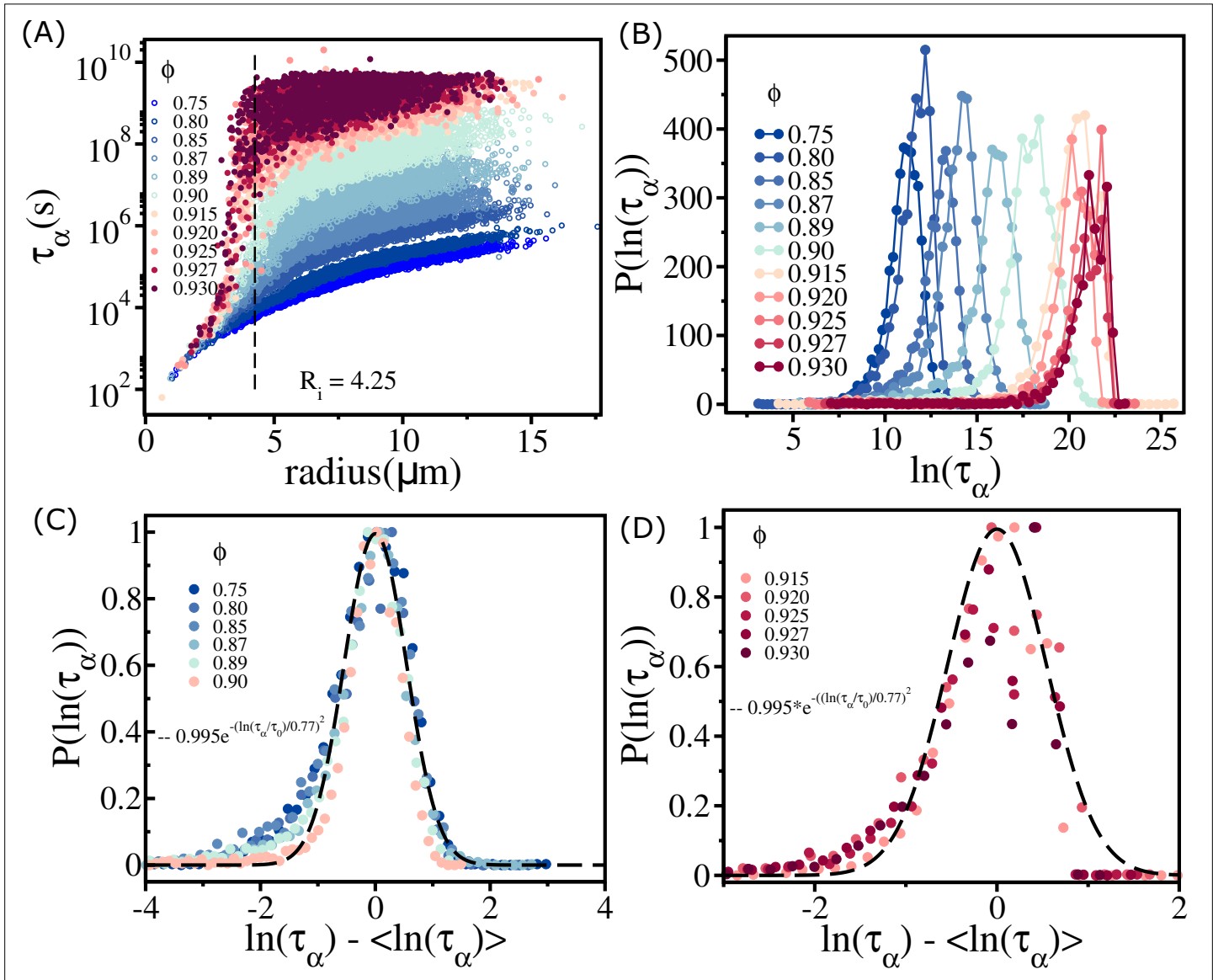

**Figure 3.** Spectrum of relaxation times. (**A**) Scatter plot of relaxation times $\tau_\alpha(s)$ as a function of cell radius. From top to bottom, the plot corresponds to decreasing $\phi$. The vertical dashed line is for $R_i = 4.25\ \mu m$, beyond which the $\tau_\alpha$ changes sharply at high packing fractions. (**B**) Histogram $P(\ln(\tau_\alpha))$ as a function of $\ln(\tau_\alpha)$. Beyond $\phi = 0.90$ ($\phi_S$), the histogram peaks do not shift substantially towards a high $\tau_\alpha$ values. (**C**) For $\phi \leq \phi_S$ (scaled by $P^{\max}(\ln(\tau_\alpha))$) falls on a master curve, as described in the main text. (**D**) Same as (**C**) except the results are for $\phi > 0.90$. The data deviates from the Gaussian fit, shown by the dashed line.

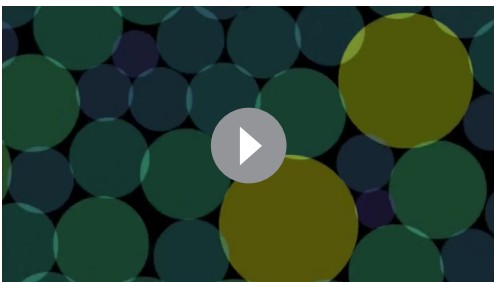

**Video 1.** Shows multiple rearrangements of smaller sized cells (blue and green cells) causes the big cells (yellow cells) to move in a highly jammed environment ($\phi = 0.92 > \phi_S$). Bright colors show the cell-cell overlap. Note that the overlap values are higher than those in lower area fractions. Free spaces (black background) are changing dynamically around a cell.
https://elifesciences.org/articles/87966/figures#video1

Simulations (*Gnan and Zaccarelli, 2019*) showed $\tau_\alpha$ increases till a critical $\phi_S$ but it decreases substantially beyond $\phi_S$ with no saturation.

## Relaxation dynamics of individual cells

Plot of $\tau_\alpha$ as a function of the radius of cells $R_i$ (*Figure 3A*) shows nearly eight orders of magnitude change. The size dependence of $\tau_\alpha$ on $\phi$ is striking. That $\tau_\alpha$ should increase for large-sized cells (see the data beyond the vertical dashed line in *Figure 3A*) is not unexpected. However, even when cell sizes increase beyond $R_i = 4.25 \ \mu m$, the dispersion in $\tau_\alpha$ is substantial, especially when $\phi$ exceeds $\phi_S$. The relaxation times for cells with $R_i < 4.25 \mu m$ are relatively short even though the system as a whole is jammed. For $\phi \geq 0.90$, $\tau_\alpha$ for

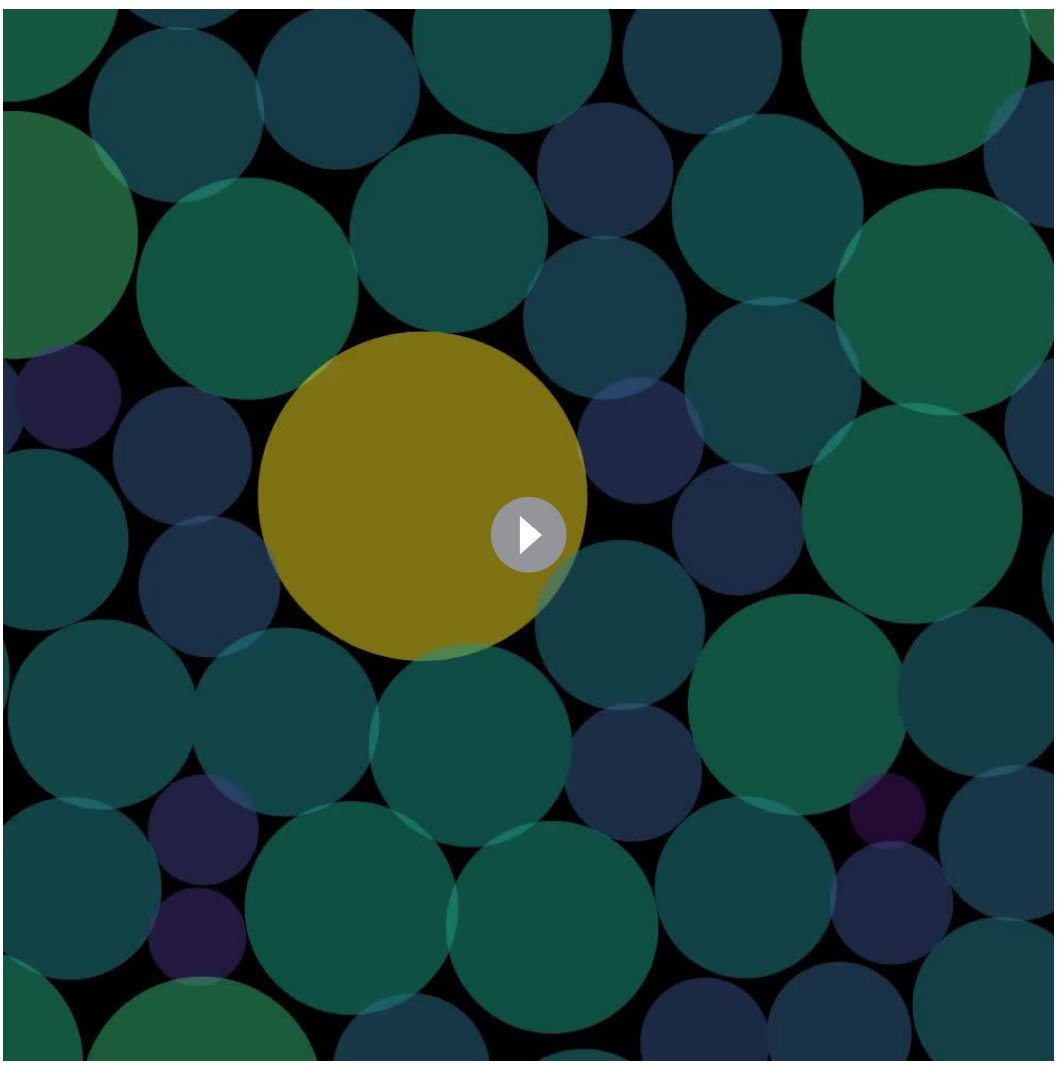

**Video 2.** Shows how a big cell (yellow) moves in the crowded environment ($\phi = 0.90(\phi_S)$). Note that the smaller-sized cells (colored as deep blue) always move faster. Again, the multiple rearrangement causes the bigger cell to move substantially. The amount of overlap is smaller than that at $\phi = 0.92$.
https://elifesciences.org/articles/87966/figures#video2

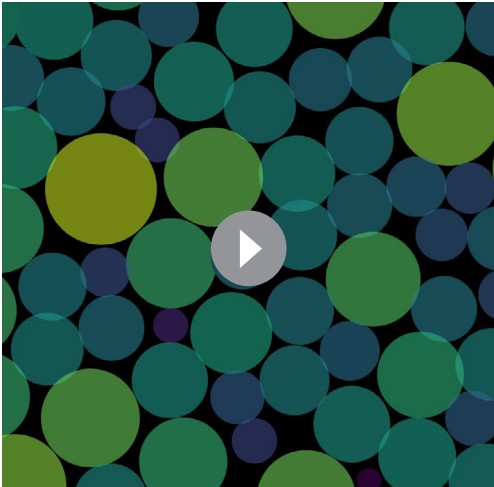

**Video 3.** Shows the movements of cells at a low area fraction ($\phi = 0.85$). Note that the smaller and bigger-sized cells are almost equally faster at lower area fractions ($phi = 0.85$) because of the huge available free areas.

https://elifesciences.org/articles/87966/figures#video3

small-sized cells have a weak dependence on $\phi$. Although $\tau_\alpha$ for cells with radius <4 μm is short, it is clear that for a given $\phi$ (e.g., $\phi = 0.93$) the variations in $\tau_\alpha$ are substantial. In contrast, $\tau_\alpha$s for larger cells ($R \geq 7\mu m$) are substantially large, possibly exceeding the typical cell division time in experiments. In what follows, we interpret these results in terms of available free area $\langle A_{\text{free}} \rangle$ for cells. The smaller-sized cells have the largest $\langle A_{\text{free}} \rangle \approx 50 \ \mu m^2 \approx \pi R_S^2 (R_S \approx 4 \ \mu m)$ ($R_S$ is the radius of the small cell).

The effect of jamming on the dramatic increase in $\tau_\alpha$ occurs near $R_i \approx 4.5 \ \mu m$, which is comparable to the length scale of short-range interactions. For $\phi \leq 0.90$, $\tau_\alpha$ increases as the cell size increases. However, at higher packing fractions, even cells of similar sizes show substantial variations in $\tau_\alpha$, which change by almost 3–4 orders of magnitude (see the data around the vertical dashed line for $\phi \geq 0.915$ in **Figure 3A**).

This is a consequence of large variations in the local density (**Appendix 6—figure 1**). Some of the similar-sized cells are trapped in the jammed environment, whereas others are in less crowded

regions (see **Appendix 6—figure 1**). The spread in $\tau_\alpha$ increases dramatically for $\phi > \phi_S (\approx 0.90)$ and effectively overlap with each other. This is vividly illustrated in the histogram, $P(\log(\tau_\alpha))$, shown in **Figure 3B**. For $\phi < \phi_s$, the peak in $P(\log(\tau_\alpha))$ monotonically shifts to higher $\log(\tau_\alpha)$ values. In contrast, when $\phi$ exceeds $\phi_S$ there is overlap in $P(\log(\tau_\alpha))$, which is reflected in the saturation of $\bar{\eta}$ and $\tau_\alpha$.

There are cells (typically with small sizes) that move faster even in a highly jammed environment (see **Appendix 5—figures 1C and 2**). The motions of the fast-moving cells change the local environment, which effectively facilitates the bigger cells to move in a crowded environment (see **Appendix 5—figures 1D and 2**, **Video 1** ($\phi = 0.92 > \phi_S$) and **Video 2** ($\phi = 0.90 = \phi_S$)). In contrast, for $\phi = 0.85 < \phi_S$, small- and large-sized cells move without hindrance because of adequate availability of free area (**Video 3**). The videos vividly illustrate the large-scale facilitated rearrangements that enable the large-sized cells to move.

The dependence of $\tau_\alpha$ on $\phi$ for $\phi \leq \phi_S$ (**Figure 2D**) implies that the polydisperse cell systems behave as a soft glass in this regime. On theoretical grounds, it was predicted that $P(\ln(\tau_\alpha)) \sim \exp[-c(\ln(\frac{\tau_\alpha}{\tau_0}))^2]$ in glass-forming systems (**Kirkpatrick and Thirumalai, 2015**). Remarkably, we found that this prediction is valid in the polydisperse cell system (**Figure 3C**). However, above $\phi_S$ the predicted relation is not satisfied (see **Figure 3D**).

## Available free area explains viscosity saturation at high $\phi$

We explain the saturation in the viscosity by calculating the available free area per cell, as $\phi$ increases. In a hard disk system, one would expect that the free area would decrease monotonically with $\phi$ until it is fully jammed at the close packing fraction (~0.84; **Drocco et al., 2005**; **Reichhardt and Reichhardt, 2014**). Because the cells are modeled as soft deformable disks, they could overlap with each other even when fully jammed. Therefore, the region where cells overlap creates free area in the immediate neighborhood.

The extent of overlap ($h_{ij}$) is reflected in distribution $P(h_{ij})$. The width in $P(h_{ij})$ increases with $\phi$, and the peak shifts to higher values of $h_{ij}$ (**Figure 4A**). The mean, $\langle h_{ij} \rangle$, increases with $\phi$ (**Figure 4B**). Thus, even if the cells are highly jammed at $\phi \approx \phi_S$, free area is available because of an increase in the overlap between cells (see **Figure 5**).

When $\phi$ exceeds $\phi_S$, the mobility of small-sized cells facilitates the larger cells to move, as is assumed in the free volume theory of polymer glasses (**Cohen and Turnbull, 1959**; **Turnbull and Cohen, 1961**; **Turnbull and Cohen, 1970**; **Falk et al., 2020**). As a result of the motion of small cells, a

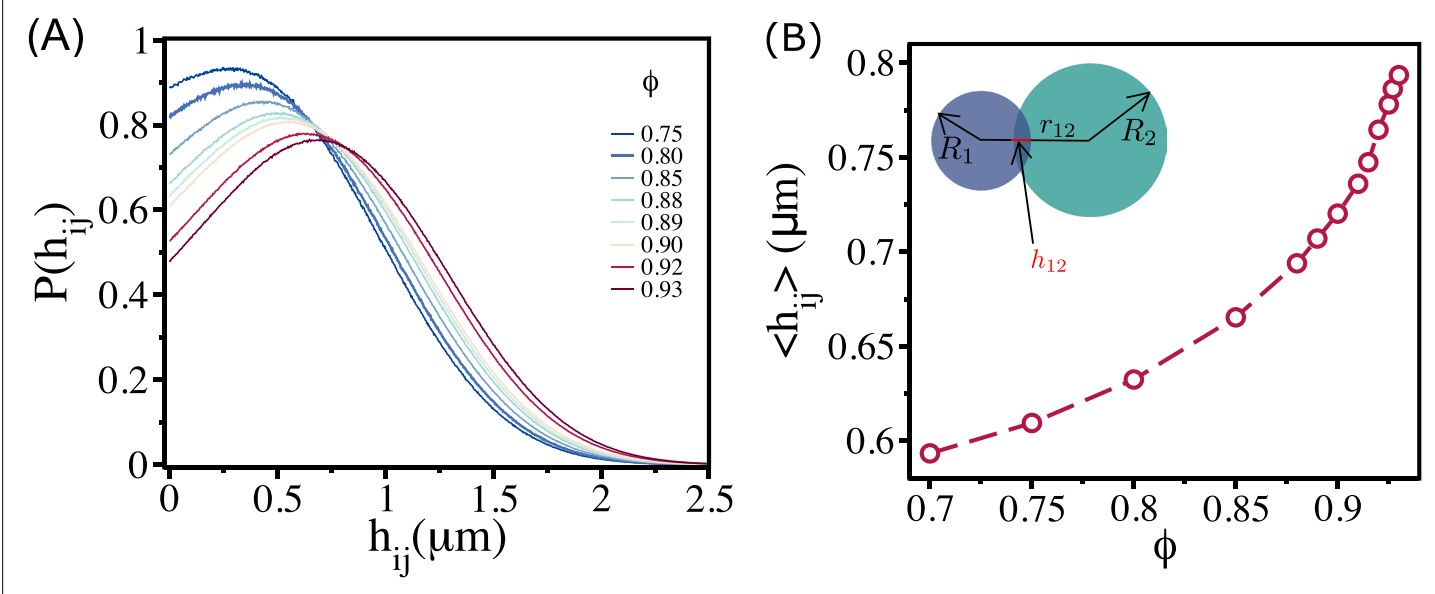

**Figure 4.** Density-dependent cell–cell overlap. (**A**) Probability of overlap ($h_{ij}$) between two cells, $P(h_{ij})$, for various $\phi$ values. The peak in the distribution function shifts to higher values as $\phi$ increases. (**B**) Mean $\langle h_{ij} \rangle = \int dh_{ij} P(h_{ij})$ as a function of $\phi$. Inset shows a pictorial illustration of $h_{12}$ between two cells with radii $R_1$ and $R_2$ at a distance $r_{12}$.

void is temporarily created, which allows other (possibly large) cells to move. In addition to the release of space, the cells can also interpenetrate (*Figure 4A and B*). If $h_{ij}$ increases, as is the case when the extent of compression increases (*Figure 4A*), the available space for nearby cells would also increase. This effect is expected to occur with high probability at $\phi_S$ and beyond, resulting in high overlap between the cells. These arguments suggest that the combined effect of PD and cell–cell overlap creates, via the self-propulsion of cells, additional free area that drives larger cells to move even under jammed conditions.

In order to quantify the physical picture given above, we calculated an effective area for each cell by first calculating Voronoi cell area $A$. A plot for Voronoi tessellation is presented in *Figure 5A* for $\phi = 0.93$, and the histogram of $A$ is shown in *Figure 5B*. As $\phi$ increases, the distribution shifts toward lower Voronoi cell size $\langle A \rangle$. The mean Voronoi cell size $\langle A \rangle$ as a function of $\phi$ in *Figure 5C* shows $\langle A \rangle$ decreases as $\phi$ is increased. As cells interpenetrate, the Voronoi cell size will be smaller than the actual cell size ($\pi R_i^2$) in many instances (*Figure 5A*). To demonstrate this quantitatively, we calculated $A_{\text{free,i}} = A_i - \pi R_i^2$. The value of $A_{\text{free}}$ could be negative if the overlap between neighboring cells is substantial; $A_{\text{free}}$ is positive only when the Voronoi cell size is greater than the actual cell size. Positive $A_{\text{free}}$ is an estimate of the available free area. The histograms of $A_{\text{free}}$ for all the packing fractions in *Figure 5D* show that the distributions saturate beyond $\phi = 0.90$. All the distributions have a substantial region in which $A_{\text{free}}$ is negative. The negative value of $A_{\text{free}}$ increases with increasing $\phi$, which implies that the amount of interpenetration between cells increases.

Because of the overlap between the cells, the available free area fraction $\phi_{\text{free}}$ is higher than the expected free area fraction ($1.0 - \phi$) for all $\phi$. We define an effective free area fraction $\phi_{\text{free}}$ as

$$\phi_{\text{free}} = \frac{\sum_{j=1}^{N_t} \sum_{i=1}^{N_p} A_{\text{free}+,i}^j}{N_t A_{\text{box}}}, \quad (4)$$

where $N_p$ is the number of positive free area in $j$th snapshots, $N_t$ is the total number of snapshots, $A_{\text{box}}$ is the simulation box area, and $A_{\text{free}+,i}^j$ is the positive free area of $i$th cell in $j$th snapshot.

The calculated $\phi_{\text{free}}$, plotted as a function of $\phi$ in *Figure 5E*, shows that $\phi_{\text{free}}$ decreases with $\phi$ until $\phi = 0.90$, and then it saturates near a value $\phi_{\text{free}} \approx 0.22$ (see the right panel in *Figure 5E*). Thus, the saturation in $\bar{\eta}$ as a function of $\phi$ is explained by the free area picture, which arises due to combined effect of the size variations and the ability of cells to overlap.

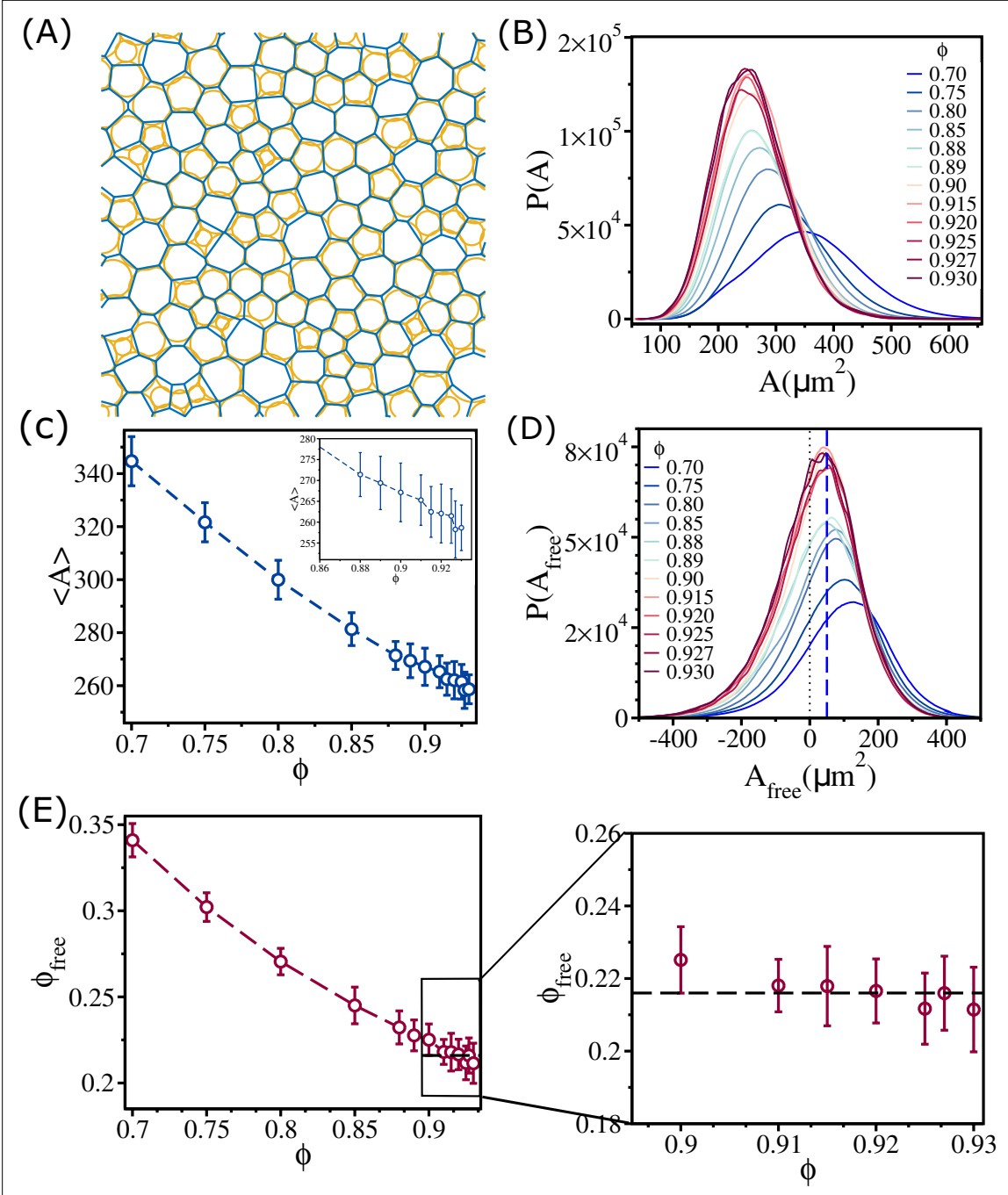

**Figure 5.** Changes in free area fraction with $\phi$. (**A**) Voronoi tessellation of cells for $\phi = 0.93$ for a single realization. The orange circles represent actual cell sizes. The blue polygons show the Voronoi cell size. (**B**) Distribution of Voronoi cell size $A$ as a function of $\phi$. (**C**) Mean Voronoi cell size $\langle A \rangle$ as a function of $\phi$. A zoomed-in view for $\phi > 0.860$ is shown in the inset. (**D**) Distribution of free area $P(A_{\text{free}})$ for all $\phi$. The vertical blue dashed line shows that the maximum in the distribution is at $A_{\text{free}} \sim 50 \mu m^2$. (**E**) Free area fraction $\phi_{\text{free}}$ as a function of $\phi$. Note that $\phi_{\text{free}}$ saturates beyond $\phi = 0.90$. An expanded view of the saturated region is shown in the right panel of (**E**). The error bars in (**C**) and (**D**) are the standard deviation in $\langle A \rangle$ and $\phi_{free}$, respectively, for 24 independent simulations.

## Aging does not explain viscosity saturation

Our main result, which we explain by adopting the free volume theory developed in the context of glasses (*Cohen and Turnbull, 1959*; *Turnbull and Cohen, 1961*; *Turnbull and Cohen, 1970*; *Falk et al., 2020*), is that above a critical packing fraction $\phi_S \sim 0.90$ the viscosity saturates. Relaxation time,

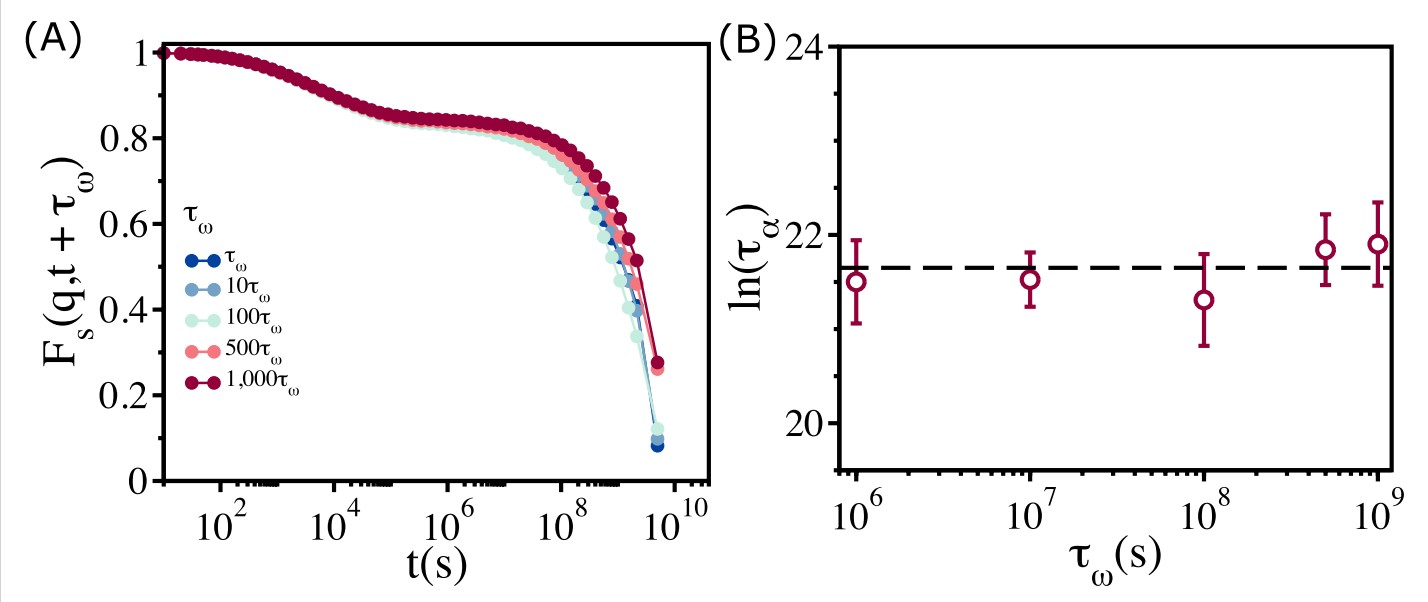

**Figure 6.** Relaxation in the polydisperse cell system is independent of the waiting time. (**A**) $F_s(q,t)$ for $\phi = 0.92$ at different waiting times ($\tau_\omega = 10^6(s)$). Regardless of the value of $\tau_\omega$, all the $F_s(q,t)$ curves collapse onto a master curve. (**B**) Relaxation time, $\ln(\tau_\alpha)$, as a function of $\tau_\omega$. Over a three orders of magnitude change in $t_\omega$, the variation in relaxation times is less than the sample-to-sample fluctuations, as shown by the error bar. The error bars in (**B**) are the standard deviation in $\tau_\alpha$ for 24 independent simulations.

$\tau_\alpha$, measured using dynamic light scattering, in nearly monodisperse microgel poly(N- isopropylacrylamide) (PNiPAM) (**Philippe et al., 2018**) was found to depend only weakly on the volume fraction (3D), if $\phi_V$ exceeds a critical value. It was suggested that the near saturation of $\tau_\alpha$ at high $\phi_V$ is due to aging, which is a non-equilibrium effect. If saturation in viscosity and relaxation time in the embryonic tissue at high $\phi$ is due to aging, then $\tau_\alpha$ should increase sharply as the waiting time, $\tau_\omega$, is lengthened. We wondered if aging could explain the observed saturation of $\eta$ in the embryonic tissue above $\phi_S$. If aging causes the plateau in the tissue dynamics, then $\eta$ or $\tau_\alpha$ should be an increasing function of the waiting time, $\tau_\omega$. To test the effect of $\tau_\omega$ on $\tau_\alpha$, we calculated the self-intermediate scattering function $F_s(q,t+\tau_\omega)$ as a function of $t$ by varying $\tau_\omega$ over three orders of magnitude at $\phi = 0.92$ (**Figure 6A**). There is literally no change in $F_s(q,t+\tau_\omega)$ over the entire range of $\tau_\omega$. We conclude that, $\tau_\alpha$, extracted from $F_s(q,t+\tau_\omega)$ is independent of $\tau_\omega$. The variations in $\tau_\alpha$ (**Figure 6B**), with respect to $\tau_\omega$, are significantly smaller than the errors in the simulation. Thus, the saturation in $\eta$ or $\tau_\alpha$ when $\phi > \phi_S$ is not a consequence of aging.

There are two implications related to the absence of aging in the dynamics of the non-confluent embryonic tissues. (i) Although active forces drive the dynamics of the cells, as they presumably do in reality, the cell collectives can be treated as being near equilibrium, justifying the use of Green–Kubo relation to calculate $\eta$. (ii) Parenthetically, we note that the absence of significant non-equilibrium effects, even though zebrafish is a living system, further justifies the use of equilibrium rigidity percolation theory to analyze the experimental data (**Petridou et al., 2021**).

## Discussion

Extensive computer simulations of a 2D dense tissue using a particle-based model of soft deformable cells with active self-propulsion have successfully reproduced the dynamical behavior observed in the blastoderm tissue of zebrafish.

The dependence of viscosity ($\eta$) and relaxation time ($\tau_\alpha$) (before the saturation) is well fit by the VFT equation. The value of $\phi_0$ obtained from simulations, $\phi_0 \sim 0.95$, is close to $\phi_0 \sim 0.94$ extracted by fitting the experimental data to the VFT equation. Thus, the dynamics for $\phi \leq \phi_S$ resembles the behavior expected for glass-forming systems. Remarkably, the dependence of $\eta$ on $\phi$ over the entire range (VFT regime followed by a plateau) may be understood using available free area picture with

essentially a single parameter, an idea that was proposed nearly 70 y ago. We discovered that PD as well as the ease of deformation of the cells that creates free area under high jamming conditions is the mechanism that explains viscosity saturation at high cell densities. The mechanism suggested here is an important step that links equilibrium PT to dynamics during zebrafish development (*Hannezo and Heisenberg, 2022*).

One could legitimately wonder if the extent of PD used in the soft discs simulations, which seems substantial, is needed to recapitulate the observed dependence of $\eta$ on $\phi$. Furthermore, such large values of PD may not represent biological tissues. Although the choice of PD was made in part by the 2D projection of area reported in experiments (*Petridou et al., 2021*), it is expected that PD values have to be less in three dimensions. We performed preliminary simulations in three dimensions with considerably reduced PD and calculated the dependence of relaxation time ($\tau_\alpha$) as a function of $\phi$. The results show that $\tau_\alpha$ does indeed saturate at high-volume fractions.

The proposed model neglects adhesive interactions between cells, which of course is not unimportant. It is crucial to wonder if the proposed mechanism would change if cell–cell adhesion is taken into account. We wanted to create the simplest model to explain the experimental data. We do think that realistic values of adhesion strength would not significantly alter the forces between cells (*Malmi-Kakkada et al., 2018*). Thus, we expect a similar mechanism. Furthermore, the physics of the dynamics in glass-forming materials does not change in systems with and without attractive forces (*Kirkpatrick and Thirumalai, 2015*). Universal behavior, such as VFT relation, is valid for a broad class of unrelated materials (see Figure 1 in *Angell, 1991*). Needless to say, non-universal quantities such as glass transition temperature $T_g$ or effective free energy barriers for relaxation will change. In our case, we expect that changing the adhesion strength, within a reasonable range, would change $\phi_S$ without qualitatively altering the dependence of $\eta$ on $\phi$. For these reasons, in the first pass we neglected adhesion, whose effects have to be investigated in the future.

In the physical considerations leading to *Equation (6)*, the random activity term ($\mu$) plays an important role. Is it possible to create a passive model by maintaining the system at a finite temperature using stochastic noise with $\mu = 0$, which would show the observed viscosity behavior? First, in such a system of stochastic equations, the coefficient of noise (a diffusion constant) would be related to $\gamma_i$ in *Equation (6)* through fluctuation dissipation theorem (FDT). Thus, only $\gamma_i$ can be varied. In contrast, in *Equation (6)* the two parameters ($\gamma_i$ and $\mu$) maybe independently changed, which implies that the two sets of stochastic equations of motion are not equivalent. Second, the passive system describes particles that interact by soft Hertz potential. In analogy with systems in which the particles interact with harmonic potential (*Ikeda et al., 2012*), we expect that the passive model would form a glass in which the viscosity would follow the VFT law.

We find it surprising that the calculation of viscosity using linear response theory (valid for systems close to equilibrium) and the link to free area quantitatively explain the simulation results and by implication the experimental data for a living and growing tissue. The calculation of free area of the cells is based on the geometrical effects of packing, which in turn is determined by cell-to-cell contact topology. These considerations, which are firmly established here, explain why equilibrium PTs are related to a steep increase in viscosity (*Kirkpatrick and Thirumalai, 2015*) as the packing fraction changes over a narrow range. The absence of aging suggests that, although a large number of cell divisions occur, they must be essentially independent, thus allowing the cells to reach local equilibrium.

## Materials and methods
### Two-dimensional cell model

Following our earlier studies (*Malmi-Kakkada et al., 2018*; *Sinha et al., 2020*), we simulated a 2D version of a particle-based cell model. We did not explicitly include cell division in the simulations. This is physically reasonable because in the experiments (*Petridou et al., 2021*) the time scales over which cell division induced local stresses relax are short compared to cell division time. Thus, local equilibrium is established in between random cell division events. We performed simulations in 2D because experiments reported the dependence of viscosity as a function of area fraction.

In our model, cells are modeled as soft deformable disks (*Matoz-Fernandez et al., 2017*; *Drasdo and Höhme, 2005*; *Schaller and Meyer-Hermann, 2005*; *Malmi-Kakkada et al., 2018*) interacting

via short-ranged forces. The elastic (repulsive) force between two cells with radii $R_i$ and $R_j$ is Hertzian, which is given by

$$F_{ij}^{el} = \frac{h_{ij}^{3/2}}{\frac{3}{2}\left(\frac{1-\nu^2}{E}\right)\sqrt{\frac{1}{R_i}+\frac{1}{R_j}}},$$

(5)

where $h_{ij} = \max[0, R_i + R_j - |\vec{r}_i - \vec{r}_j|]$. The repulsive force acts along the unit vector $\vec{n}_{ij}$, which points from the center of the $j$th cell to the center of the $i$th cell. The total force on the $i$th cell is

$$\vec{F}_i = \sum_{j\in NN(i)} \left(F_{ij}^{el}\right)\vec{n}_{ij},$$

where $NN(i)$ is the number of near-neighbor cells that are in contact with the $i$th cell. The $j$th cell is the nearest neighbor of the $i$th cell, if $h_{ij} > 0$. The near-neighbor condition ensures that the cells interpenetrate each other to some extent, thus mimicking the cell softness. For simplicity, we assume that the elastic moduli ($E$) and the Poisson ratios ($\nu$) for all the cells are identical. PD in the cell sizes is important in recovering the plateau in the viscosity as a function of packing fraction. Thus, the distribution of cell areas ($A_i = \pi R_i^2$) is assumed to have a distribution that mimics the broad area distribution discovered in experiments.

## Self-propulsion and equations of motion

In addition to the repulsive Hertz force, we include an active force arising from self-propulsion mobility ($\mu$), which is a proxy for the intrinsically generated forces within a cell. For illustration purposes, we take $\mu$ to independent of the cells, although this can be relaxed readily. We assume that the dynamics of each cell obeys the phenomenological equation

$$\dot{\vec{r}}_i = \frac{\vec{F}_i}{\gamma_i} + \mu\vec{\mathcal{W}}_i(t),$$

(6)

where $\gamma_i$ is the friction coefficient of $i$th cell, and $\mathcal{W}_i(t)$ is a noise term. The friction coefficient $\gamma_i$ is taken to be $\gamma_0 R_i$ (*Sinha et al., 2022*). By scaling $t$ by the characteristic time scale, $\tau = \frac{\langle R\rangle^2}{\mu^2}$ in *Equation (6)*, one can show that the results should be insensitive to the exact value of $\mu$. The noise term $\mathcal{W}_i(t)$ is chosen such that $\langle\mathcal{W}_i(t)\rangle = 0$ and $\langle\mathcal{W}_i^\alpha(t)\mathcal{W}_j^\beta(t')\rangle = \delta(t-t')\delta_{i,j}\delta^{\alpha,\beta}$. In our model, there is no dynamics with only systematic forces because the temperature is zero. The observed dynamics arises solely due to the self-propulsion (*Equation 6*).

We place $N$ cells in a square box that is periodically replicated. The size of the box is $L$ so that the packing fraction (in our 2D system it is the area fraction) is $\phi = \frac{\sum_{i=1}^{N}\pi R_i^2}{L^2}$. We performed extensive simulations by varying $\phi$ in the range $0.700 \leq \phi \leq 0.950$. The results reported in main text are obtained with $N = 500$. Finite size effects are discussed in Appendix 7.

To mimic the variations in the area of cells in a tissue (*Petridou et al., 2021*), we use a broad distribution of cell radii (see Appendix 1 for details). The parameters for the model are given in *Table 1*. In this study, we do not consider the growth and division of cells. Thus, our simulations describe

**Table 1.** Parameters used in the simulation.

| Parameters | Values | References |
|---|---|---|
| Timestep ($\Delta t$) | 10s | This paper |
| Self-propulsion ($\mu$) | $0.045\mu$m/$\sqrt{s}$ | This paper |
| Friction coefficient ($\gamma_o$) | 0.1kg/($\mu$m s) | This paper |
| Mean cell elastic modulus ($E_i$) | $10^{-3}$MPa | *Galle et al., 2005*; *Malmi-Kakkada et al., 2018* |
| Mean cell Poisson ratio ($\nu_i$) | 0.5 | *Schaller and Meyer-Hermann, 2005*; *Malmi-Kakkada et al., 2018* |

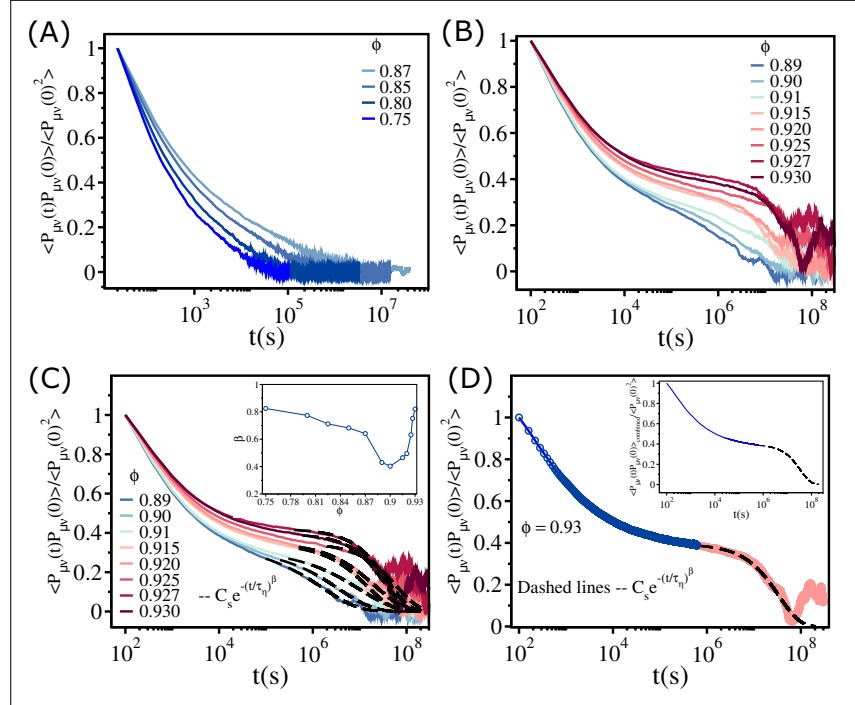

**Figure 7.** Fit of the stress–stress correlation functions to stretched exponential functions. (**A**) The stress–stress correlation function $\langle P_{\mu\nu}(t)P_{\mu\nu}(0)\rangle$ divided by the value at $t = 0$ as a function of $t$ for $\phi \in (0.75 - 0.87)$. (**B**) Similar plot for $\phi \in (0.89 - 0.93)$. (**C**) The long time decay of $\langle P_{\mu\nu}(t)P_{\mu\nu}(0)\rangle$ is fit to $C_s \exp\left[-\left(\frac{t}{\tau_\eta}\right)^\beta\right]$, as shown by the dashed lines. The inset shows the dependence of $\beta$ on $\phi$. (**D**) The data that is fit using the stretched exponential function (black dashed line) is combined with the short time data (blue solid line), which is fit using the cubic spline function. The resulting fits produces a smooth curve $\langle P_{\mu\nu}(t)P_{\mu\nu}(0)\rangle_{\text{combined}}$, as shown in the inset.

steady-state dynamics of the tissue. For each $\phi$, we performed simulations for at least $(5-10)\tau_\alpha$ before storing the data. For each $\phi$, we performed 24 independent simulations. The calculation of viscosity was performed by averaging over 40 independent simulations at each $\phi$.

## Calculation of viscosity

We calculated the effective viscosity ($\bar{\eta}$) for various values of $\phi$ by integrating the off-diagonal part of the stress–stress correlation function $\langle P_{\mu\nu}(t)P_{\mu\nu}(0)\rangle$ using the Green–Kubo relation (**Hansen and McDonald, 2013**) (without the pre-factor $\frac{A}{k_BT}$)

$$\bar{\eta} = \int_0^\infty dt \sum_{(\mu\nu)} \langle P_{\mu\nu}(t)P_{\mu\nu}(0)\rangle, \tag{7}$$

where $\mu$ and $\nu$ denote Cartesian components ($x$ and $y$) of the stress tensor $P_{\mu\nu}(t)$ (see main text for the definition of $P_{\mu\nu}(t)$). The definition of $\bar{\eta}$, which relates the decay of stresses as a function of times in the non-confluent tissue, is akin to the methods used to calculate viscosity in simple fluids (**Equation 7**). The time dependence of $\langle P_{\mu\nu}(t)P_{\mu\nu}(0)\rangle$, normalized by $\langle P_{\mu\nu}(0)^2\rangle$, for different values of $\phi$ (**Figure 7A and B**) shows that the stress relaxation is clearly non-exponential, decaying to zero in two steps. After an initial rapid decay followed by a plateau at intermediate times (clearly visible for $\phi \geq 0.91$), the normalized $\langle P_{\mu\nu}(t)P_{\mu\nu}(0)\rangle$ decays to zero as a stretched exponential. The black dashed lines in **Figure 7C** show that a stretched exponential function, $C_s \exp\left[-\left(\frac{t}{\tau_\eta}\right)^\beta\right]$, where $\tau_\eta$ is the characteristic time in which stress relax and $\beta$ is the stretching exponent, provides an excellent fit to the long time decay of $\langle P_{\mu\nu}(t)P_{\mu\nu}(0)\rangle$ (from the plateau region to zero) as a function of $t$. Therefore, we utilized the fit function, $C_s \exp\left[-\left(\frac{t}{\tau_\eta}\right)^\beta\right]$, to replace the noisy long time part of $\langle P_{\mu\nu}(t)P_{\mu\nu}(0)\rangle$ by a smooth fit

data before evaluating the integral in *Equation (7)*. The details of the procedure to compute $\bar{\eta}$ are described below.

We divided $\langle P_{\mu\nu}(t)P_{\mu\nu}(0)\rangle$ in two parts. (i) The short time part ($\langle P_{\mu\nu}(t)P_{\mu\nu}(0)\rangle_{\text{short}}$) – the smooth initial rapid decay until the plateau is reached (e.g., see the blue circles in *Figure 7D* for $\phi = 0.93$). For the $n$ data points at short times, $\left(t_1, \langle P_{\mu\nu}(t_1)P_{\mu\nu}(0)\rangle_{\text{short}}\right),\ldots,\left(t_n, \langle P_{\mu\nu}(t_n)P_{\mu\nu}(0)\rangle_{\text{short}}\right)$, we constructed a spline $S(t)$ using a set of cubic polynomials:

$$
\begin{aligned}
S_1(t) &= \langle P_{\mu\nu}(t_1)P_{\mu\nu}(0)\rangle_{\text{short}} + b_1(t - t_1) + c_1(t - t_1)^2 + d_1(t - t_1)^3 \\
S_2(t) &= \langle P_{\mu\nu}(t_2)P_{\mu\nu}(0)\rangle_{\text{short}} + b_2(t - t_2) + c_2(t - t_2)^2 + d_2(t - t_2)^3 \\
S_{n-1}(t) &= \langle P_{\mu\nu}(t_{n-1})P_{\mu\nu}(0)\rangle_{\text{short}} + b_{n-1}(t - t_{n-1}) + c_{n-1}(t - t_{n-1})^2 + d_{n-1}(t - t_{n-1})^3.
\end{aligned}
$$

The polynomials satisfy the following properties. (a) $S_i(t_i) = \langle P_{\mu\nu}(t_i)P_{\mu\nu}(0)\rangle_{\text{short}}$ and $S_i(t_{i+1}) = \langle P_{\mu\nu}(t_{i+1})P_{\mu\nu}(0)\rangle_{\text{short}}$ for $i = 1, \ldots, n-1$ which guarantees that the spline function $S(t)$ interpolates between the data points. (b) $S'_{i-1}(t) = S'_i(t)$ for $i = 2, \ldots, n-1$ so that $S'(t)$ is continuous in the interval $[t_1, t_n]$. (c) $S''_{i-1}(t) = S''_i(t)$ for $i = 2, \ldots, n-1$ so that $S''(t)$ is continuous in the interval $[t_1, t_n]$. By solving for the unknown parameters, $b_i, c_i$, and $d_i$, using the above-mentioned properties, we constructed the function $S(t)$. We used $S(t)$ to fit $\langle P_{\mu\nu}(t)P_{\mu\nu}(0)\rangle_{\text{short}}$ to get an evenly spaced ($\delta t = 10s$) smooth data (solid blue line in *Figure 7D*). The fitting was done using the software 'Xmgrace'.

(ii) The long time part ($\langle P_{\mu\nu}(t)P_{\mu\nu}(0)\rangle_{\text{long}}$) – from the plateau until it decays to zero – is shown by the red circles in *Figure 7D*. The long time part was fit using the analytical function $C_s \exp\left[-\left(\frac{t}{\tau_\eta}\right)^\beta\right]$ (black dashed line in *Figure 7D*). We refer to the fit data ($\delta t = 10s$) as $\langle P_{\mu\nu}(t)P_{\mu\nu}(0)\rangle_{\text{long}}^{\text{fit}}$.

We then combined $\langle P_{\mu\nu}(t)P_{\mu\nu}(0)\rangle_{\text{short}}$ and $\langle P_{\mu\nu}(t)P_{\mu\nu}(0)\rangle_{\text{long}}^{\text{fitted}}$ to obtain $\langle P_{\mu\nu}(t)P_{\mu\nu}(0)\rangle_{\text{combined}}$ (see inset of *Figure 7D*). Finally, we calculated $\bar{\eta}$ using the equation,

$$
\begin{aligned}
\bar{\eta} &= \lim_{\delta t \to 0} \sum_{i=0}^{T} \delta t \sum_{(\mu\nu)} \langle P_{\mu\nu}(i\delta t)P_{\mu\nu}(0)\rangle_{\text{combined}} \\
&= \lim_{\delta t \to 0} \sum_{i=0}^{t_1} \delta t \sum_{(\mu\nu)} \langle P_{\mu\nu}(i\delta t)P_{\mu\nu}(0)\rangle_{\text{short}} \\
&\quad + \lim_{\delta t \to 0} \sum_{i=t_1}^{T} \delta t \sum_{(\mu\nu)} \langle P_{\mu\nu}(i\delta t)P_{\mu\nu}(0)\rangle_{\text{long}}^{\text{fit}},
\end{aligned}
\tag{8}
$$

where $t_1 \delta t$ is the end point of $\langle P_{\mu\nu}(t)P_{\mu\nu}(0)\rangle_{\text{short}}$ and $T\delta t$ is the end point of $\langle P_{\mu\nu}(i\delta t)P_{\mu\nu}(0)\rangle_{\text{combined}}$.

## Acknowledgements

We acknowledge Anne D Bowen at the Visualization Laboratory (Vislab), Texas Advanced Computing Center, for help with video visualizations. We are grateful to Mauro Mugnai and Nicoletta Petridou for useful discussions. This work was supported by grants from the National Science Foundation (PHY 23-10639 and CHE 23-20256) and the Welch Foundation (F-0019).

## Additional information

### Funding

| Funder | Grant reference number | Author |
| --- | --- | --- |
| National Science Foundation | 2345242 | D Thirumalai |
| Welch Foundation | F-0019 | D Thirumalai |

The funders had no role in study design, data collection and interpretation, or the decision to submit the work for publication.

### Author contributions

Rajsekhar Das, Conceptualization, Resources, Data curation, Software, Formal analysis, Investigation, Visualization, Writing – original draft, Writing – review and editing; Sumit Sinha, Conceptualization, Formal analysis, Investigation, Methodology, Writing – original draft, Writing – review and editing; Xin Li, Conceptualization, Formal analysis, Writing – original draft, Writing – review and editing; TR

Kirkpatrick, Conceptualization, Formal analysis, Methodology, Writing – original draft; D Thirumalai, Conceptualization, Formal analysis, Supervision, Funding acquisition, Investigation, Methodology, Writing – original draft, Project administration, Writing – review and editing

**Author ORCIDs**
Rajsekhar Das (ID) http://orcid.org/0000-0003-2626-7259
Sumit Sinha (ID) http://orcid.org/0000-0002-8364-5175
Xin Li (ID) http://orcid.org/0000-0002-2510-2236
D Thirumalai (ID) https://orcid.org/0000-0003-1801-5924

Joint Public Review: https://doi.org/10.7554/eLife.87966.4.sa1
Author Response https://doi.org/10.7554/eLife.87966.4.sa2

---

## Additional files

**Supplementary files**
• MDAR checklist

**Data availability**
The experimental data was extracted from *Petridou et al., 2021*, and the data is available in GitHub link: https://github.com/rajsekhardas88/eLife (copy archived at *Rajsekhar, 2023*).

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

## Appendix 1

## Cell polydispersity is needed to account for viscosity saturation

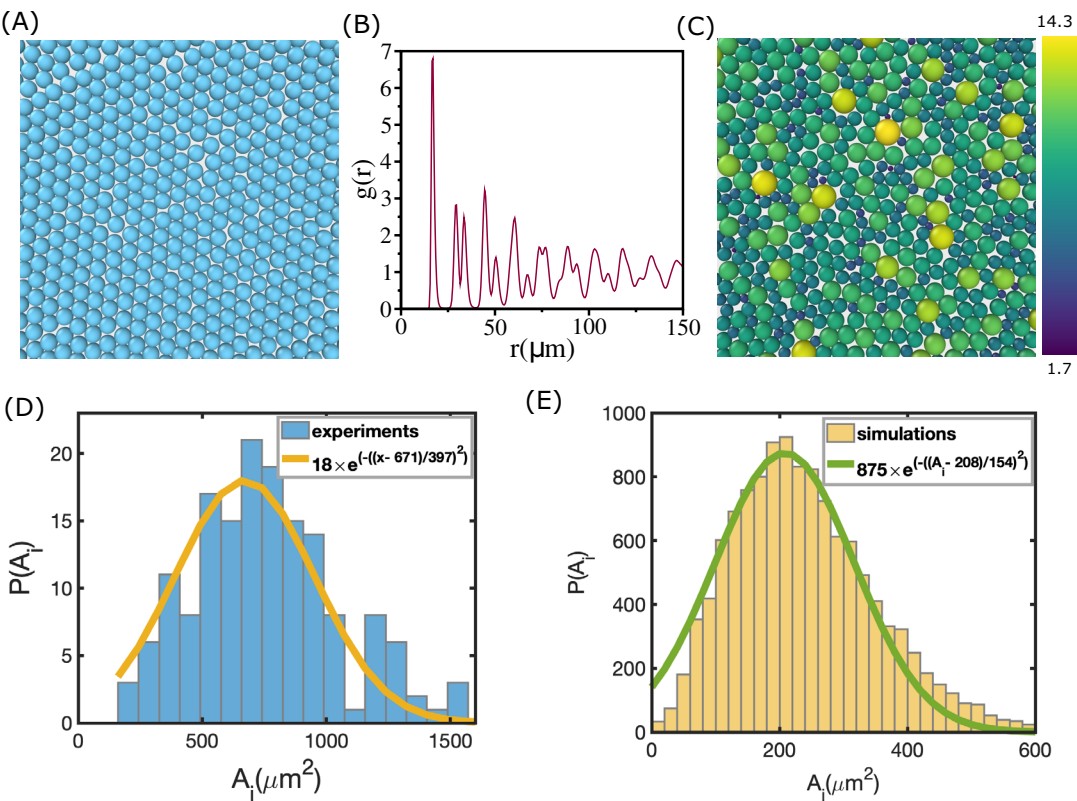

**Appendix 1—figure 1.** Area distribution of the cells. (**A**) Simulation snapshot for monodisperse cell system. The number of cells in the two-dimensional periodic box is $N = 500$. (**B**) Pair correlation function, $g(r)$, as a function of $r$. There is clear evidence of order, as reflected in the sharp peaks at regular intervals, which reflects the packing in (**A**). (**C**) A schematic picture of polydisperse cell system from the simulations. Color bar on the right shows the scale of radii in μm. There is no discernible order. (**D**) Distribution of cell area extracted from experiment during morphogenesis of zebrafish blastoderm (extracted from Fig. S2(A)) (*Petridou et al., 2021*). (**E**) Same as (**D**) except, $P(A_i)$, used in a typical simulation. Cell radii vary from 2 μm to 15 μm.

To explain the observed saturation in viscosity (*Figure 1C*) when cell area fraction, $\phi$, exceeds $\phi_S$ ($\approx 0.90$), we first simulated a monodisperse cell system using the model described in the 'Materials and methods' section. The monodisperse cell system ($R = 8.5\mu m$) crystallizes (see *Appendix 1— figure 1A and B*), which excludes it from being a viable model for explaining the experimental findings. We also find that a system consisting of 50:50 binary mixture of cells cannot account for the experimental data even though crystallization is avoided (see the next section). These findings forced us to take PD into account.

In order to develop an agent-based model that accounts for PD effects, we first extracted the distribution, $P(A_i)$, of the area ($A_i$) in the zebrafish cells from the experimental data (*Petridou et al., 2021*). *Appendix 1—figure 1D* shows that $P(A_i)$ is broad, implying that cell sizes are highly heterogeneous. Based on this finding, we sought a model of the non-confluent tissue that approximately mimics PD found in experiments. In other words, $\frac{\Delta A}{\langle A \rangle}$ ($\Delta A$ is the dispersion in $P(A_i)$ and $\langle A \rangle$ is the mean value) should be similar to the data in *Appendix 1—figure 1D*. The radii ($R_i$) of the cells in the simulations are sampled from a Gaussian distribution $\sim \exp\left(-(R_i - \langle R \rangle)^2/2\Delta R^2\right)$, with $\langle R \rangle = 8.5\mu m$ and $\Delta R = 4.5\mu m$. The resulting $P(A_i)$ for one of the realizations (see *Appendix 1—figure 1C*) is shown in *Appendix 1—figure 1E*. The value of $\frac{\Delta A}{\langle A \rangle}$ is $\sim 0.5$, which compares favorably with the experimental estimate ($\sim 0.4$). The unusual dependence of the viscosity ($\eta$) as a function of packing fraction ($\phi$) cannot be reproduced in the absence of PD.

# Appendix 2

## Relaxation time in the binary mixture of cells does not saturate at high $\phi$

We showed that the saturation in the relaxation time above a critical area fraction, $\phi_S$, is related to two factors. One is that there ought to be dispersion in the cell sizes (*Appendix 1—figure 1*). The extent of dispersion is likely to less in three rather in two dimensions. The second criterion is that the cells should be soft, allowing them to overlap at high area fraction. In other words, the cell diameters are explicitly non-additive. The Hertz potential captures the squishy nature of the cells.

In order to reveal the importance of PD, we first investigated if a binary mixture of cells (see *Appendix 2—figure 1A*) would reproduce the observed dependence of $\tau_\alpha$ on $\phi$. We created a 50:50 mixture with a cell size ratio $\sim 1.4$ ($R_B = 8.5\mu m$ and $R_S = 6.1\mu m$). All other parameters are the same as in the polydisperse system (see *Table 1*). The radial distribution function, $g(r)$, calculated by considering both the cell types, exhibits intermediate- but not long-range translational order (*Appendix 2—figure 1B*). A peak in $g(r)$ appears at $r_{max} = 14.0\mu m$.

In order to determine the dependence of the relaxation time, $\tau_\alpha$, on $\phi$, we first calculated $F_s(q,t)$ at $q = 2\pi/r_{max}$ (*Appendix 2—figure 1C*). The decay of $F_s(q,t)$ is similar to what one finds in typical glass-forming systems.

As $\phi$ increases, the decay of $F_s(q,t)$ slows down dramatically. When $\phi$ exceeds 0.93, there is a visible plateau at intermediate times followed by a slow decay. By fitting the long time decay of $F_s(q,t)$ to $\exp(-(t/\tau_\alpha)^\beta)$ ($\beta$ is the stretching exponent), we find that $\tau_\alpha(\phi)$ as a function of $\phi$ is well characterized by the VFT relation (*Appendix 2—figure 1D*). There is no evidence of saturation in $\tau_\alpha(\phi)$ at high $\phi$. The $\phi$ dependence of the effective shear viscosity $\bar{\eta}$ (*Appendix 2—figure 1E*), calculated using *Equation (8)*, also shows no sign of saturation. The VFT behavior, with $\phi_0 \approx 1$ and $D \approx 1.2$, shows that the two-component cell system behaves as a 'fragile' glass (*Angell, 1991*).

A comment regarding the binary system of cells is in order. The variables that characterize this system are $\lambda = R_B/R_S$ ($R_B$ ($R_S$) is the radius of the big (small) cells), $\Phi_B = N_B/(N_A + N_B)$ ($N_B$ ($N_A$) is the number of big (small) cells) with $\Phi_A = 1 - \Phi_B$, and the packing fraction, $\phi = \pi(N_B R_B^2 + N_A R_A^2)/A_S$, where $A_S$ is the area of the sample. The results in *Appendix 2—figure 1* were obtained using $\lambda = 1.4$, $\Phi_B = 0.5$. With this choice, the value of PD is $(\lambda - 1)/(\lambda + 1) \approx 0.17$, which is smaller than the experimental value. It is possible that by thoroughly exploring the parameter space, $\lambda$ and $\phi$, one could find regions in which the $\tau_\alpha$ in the two-component cell system would saturate beyond $\phi_S$. However, in light of the results in *Appendix 1—figure 1D*, we choose to simulate systems that also have high degree of PD (*Appendix 1—figure 1E*).

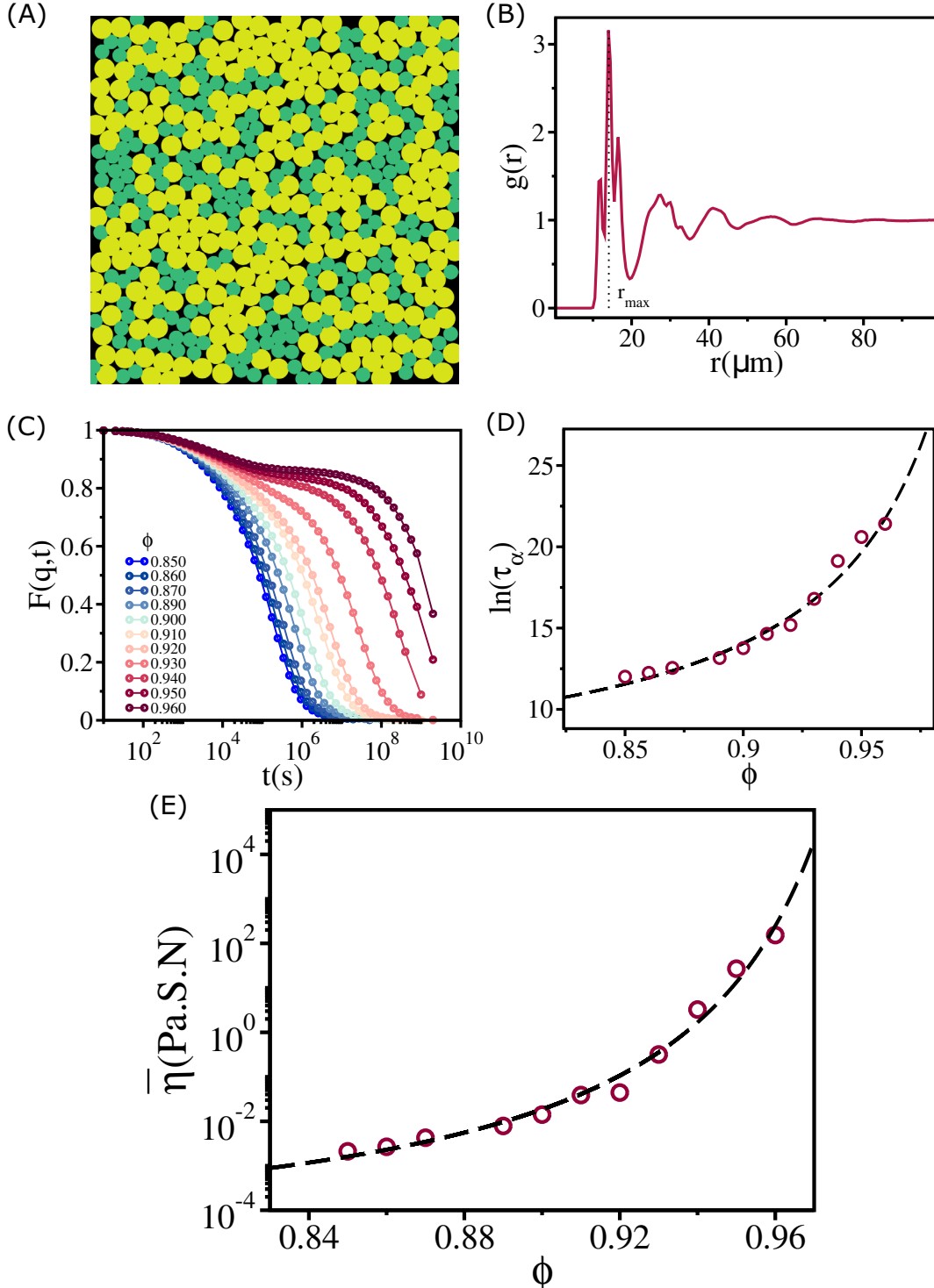

**Appendix 2—figure 1.** Structure and relaxation behavior for a binary mixture of cells. (**A**) A typical simulation snapshot for binary mixture of cells at $\phi = 0.93$. (**B**) The corresponding pair correlation function, $g(r)$, between all the cells. The vertical dashed line is at the first peak position ($r_{max}$). (**C**) $F_s(q,t)$, with $q = \frac{2\pi}{r_{max}}$, where $r_{max}$ is the location of the first peak in the $g(r)$, as a function of time at various $\phi$ values. (**D**) The logarithm of the relaxation time, $\tau_\alpha$, as a function of $\phi$. Over the entire range of $\phi$, the increase in $\tau_\alpha$ is well fit by the Vogel–Fulcher–Tammann (VFT) (VFT) relation. Most importantly, the relaxation time does not saturate, which means the evolving tissue cannot be modeled using a 50:50 binary mixture. (**E**) Effective shear viscosity $\bar{\eta}$ as a function of $\phi$ reflects the behavior of $\tau$ as a function of $\phi$ in (**D**).

## Appendix 3

### Absence of saturation in the free area in binary mixture of cells

In the main text, we established that the effective viscosity, $\bar{\eta}$, which should be a proxy for the true $\eta$ and the relaxation time ($\tau_\alpha$) in the polydisperse cell system, saturates beyond $\phi_S$. The dynamics saturates beyond $\phi_S$ because the available area per cells (quantified as $\phi_{\text{free}}$) is roughly a constant in this region (see *Figure 5E*). In the binary system, however, we do not find any saturation in the $\tau_\alpha$. Therefore, if $\phi_{\text{free}}$ controls the dynamics, one would expect that $\phi_{\text{free}}$ should decrease monotonically with $\phi$ for the binary cells system. We calculated the average Vornoi cell size $\langle A \rangle$ (*Appendix 3—figure 1A*) and $\phi_{\text{free}}$ (*Appendix 3—figure 1B*) for the binary system. Both $\langle A \rangle$ and $\phi_{\text{free}}$ decrease with $\phi$, which is consistent with the free volume picture proposed in the context of glasses.

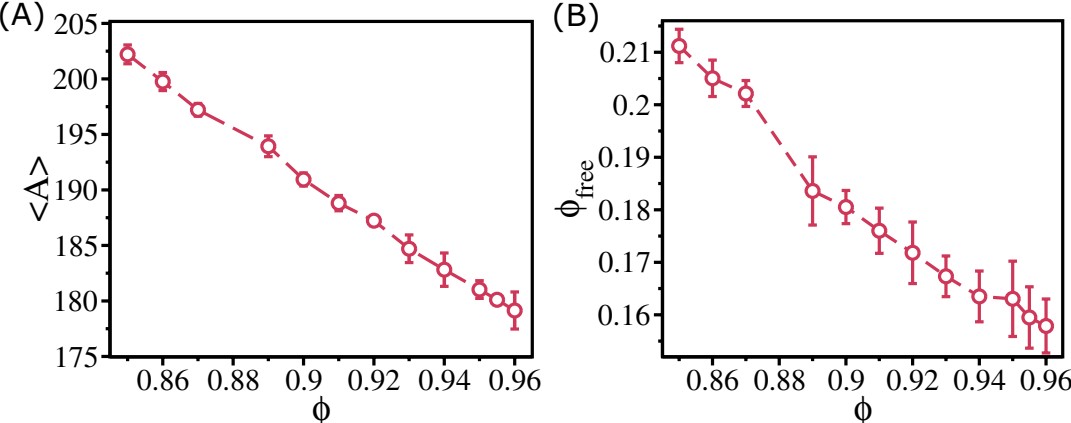

**Appendix 3—figure 1.** Free area decreases monotonically for the binary mixture of cells. (**A**) Mean Voronoi cell size, $\langle A \rangle$, as a function of $\phi$ for the 50:50 binary system. (**B**) The free area fraction, $\phi_{\text{free}}$, as a function of $\phi$ shows that $\phi_{\text{free}}$ decreases monotonically as $\phi$ increases. The error bars in (**B**) are the standard deviation in $\phi_{free}$ for 24 independent simulations.

## Appendix 4

### Absence of broken ergodicity

We have shown in the earlier section that the saturation of $\tau_\alpha$ above $\phi_S$ is not a consequence of aging. In the context of glasses and supercooled liquids (*Thirumalai et al., 1989*), it has been shown that if ergodicity is broken, then a variety of observables would depend on initial conditions. In order to test if ergodicity is broken in our model of non-confluent tissues, we define a measure $\Omega(t)$ given by

$$\Omega(t) = \left[\omega^k(t) - \omega^l(t)\right]^2, \tag{8}$$

where $k$ and $l$ represent systems with different initial conditions, and $\omega^k(t) = \frac{1}{t}\int_0^t P_{\mu\nu}^k(s)ds$ with $P_{\mu\nu}(s)$ being the value of stress (see *Equation 2* in the main text) at time $s$. If the system is ergodic, implying the system has explored the whole phase space on the simulation time scale, the values of $\omega^k(t)$ and $\omega^l(t)$ would be independent of $k$ and $l$. Therefore, $\Omega(t)$ should vanish at very long time for ergodic systems. On the other hand, if ergodicity is broken, then $\Omega(t)$ would be a constant whose value would depend on $k$ and $l$.

The long time values of $\Omega(t)$, normalized by $\Omega(0)$ (at $t=0$), are 0.01, 0.016, and 0.026 for $\phi = 0.85, 0.90$, and 0.92, respectively (*Appendix 4—figure 1A–C*). Because these values are sufficiently small, we surmise that effectively ergodicity is established. Therefore, our conclusion in the earlier section that the polydisperse cell system is in near equilibrium is justified and also explains the absence of aging. Furthermore, it was also predicted previously (*Thirumalai et al., 1989*) that in long time the ergodic measure ($\Omega(t)$ in our case) should decay as $\approx 1/t$. *Appendix 4—figure 1D* shows that this is indeed the case – at long time $\Omega(t)/\Omega(0)$ decays approximately as $1/t$.

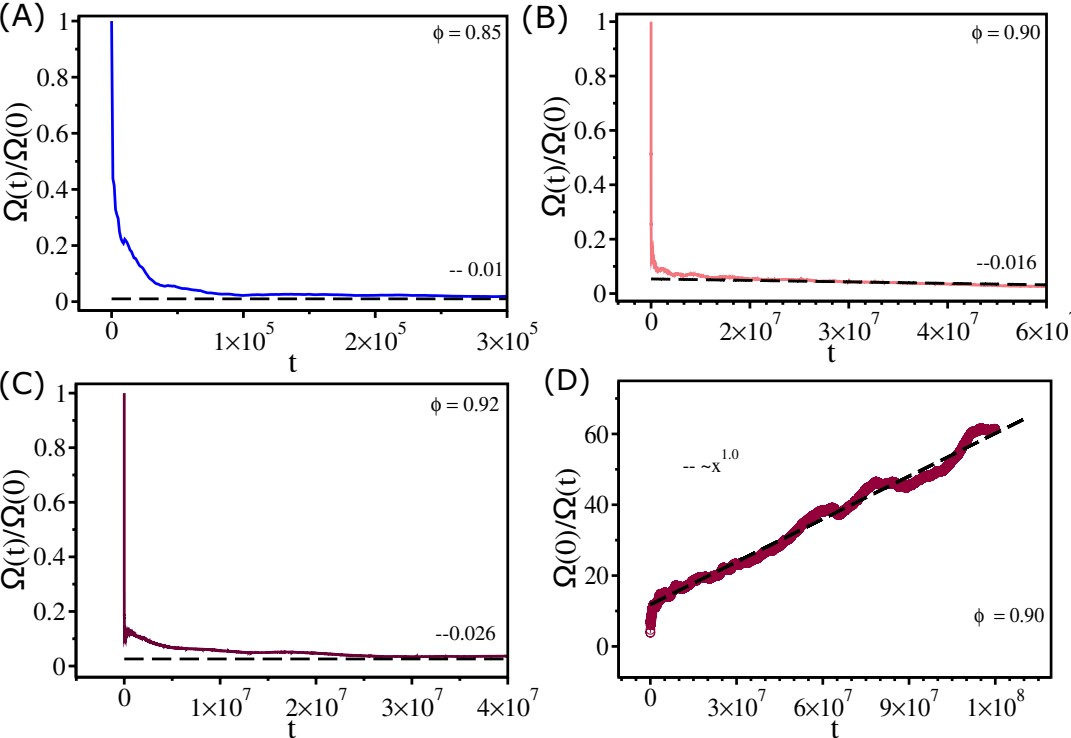

**Appendix 4—figure 1.** Measure of ergodicity. (**A**) Ergodic measure $\Omega(t)$ scaled by the value at $t=0$ ($\Omega(0)$) as a function of $t$ for $\phi = 0.85$. (**B, C**) Similar plots for $\phi = 0.90$ and $\phi = 0.92$, respectively. At long time, $\Omega(t)/\Omega(0)$ reach 0.01, 0.016, and 0.026 for $\phi = 0.85$, $\phi = 0.90$, and $\phi = 0.92$, respectively. (**D**) $\Omega(0)/\Omega(t)$ as a function of $t$ for $\phi = 0.90$. The dashed line shows a linear fit. The time t is in second.

## Appendix 5

### Dynamics of small and large cells are dramatically different

The structural and the dynamical behavior of the small ($R_S \leq 4.5 \mu m$) and large ($R_B \geq 12.0 \mu m$) cells is dramatically different in the non-confluent tissue. The pair correlation functions between small cells ($g_{SS}(r)$) and between large cells ($g_{BB}(r)$) (**Appendix 5—figure 1A and B**) for $\phi = 0.905$ and $\phi = 0.92$ show that both small and large cells exhibit liquid-like disordered structures. However, it is important to note that $g_{SS}(r)$ has only one prominent peak and a modest second peak. In contrast, $g_{BB}(r)$ has three prominent peaks. Thus, the smaller-sized cells exhibit liquid-like behavior, whereas the large cells are jammed. This structural feature is reflected in the decay of $F_s(q,t)$ with $q = \frac{2\pi}{r_{max}}$, where $r_{max}$ is the position of the first peak in the $g(r)$ (**Appendix 5—figure 1C and D**).

There is a clear difference in the decay of $F_s(q,t)$, spanning nearly eight orders of magnitude, between small and large cells (compare **Appendix 5—figure 1C and D**). At the highly jammed packing fraction ($\phi = 0.92$), small-sized cells have the characteristics of fluid-like behavior.

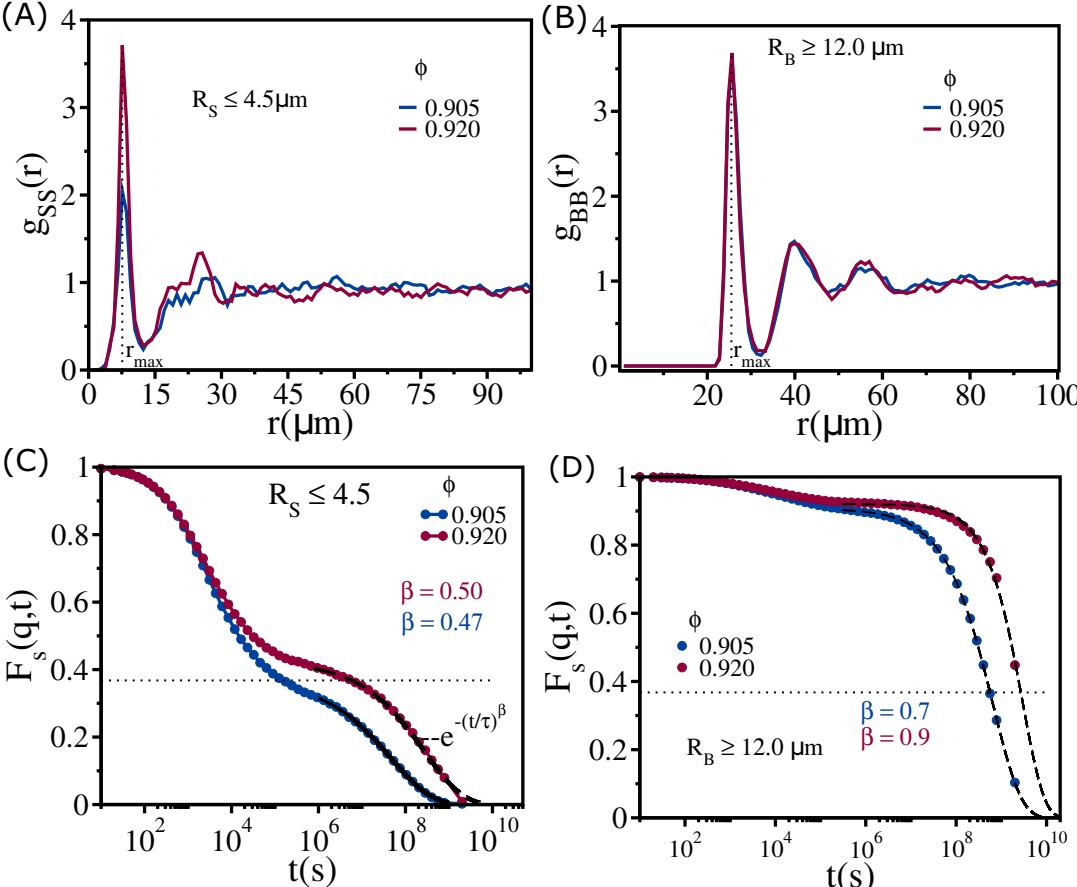

**Appendix 5—figure 1.** Cell size-dependent structures and dynamics. (**A**) Radial distribution function $g_{SS}(r)$ between small-sized cells ($R_S \leq 4.5 \mu m$) at $\phi = 0.905$ (blue) and $0.92$ (red). These values are greater than $\phi_s \approx 0.90$. (**B**) Same as (**A**) except the results are for $g_{BB}(r)$ between large cells ($R_B \geq 12.0 \mu m$). (**C**) $F_s(q,t)$ for cells with $R_S \leq 4.5 \mu m$ at $\phi = 0.905$ and $\phi = 0.92$. Note that even at these dense packings, the mobility of the smaller-sized cells is substantial, which is reflected in the time dependence of $F_s(q,t)$. (**D**) $F_s(q,t)$ for cells with $R_B \geq 12.0 \mu m$ at $\phi = 0.905$ and $\phi = 0.92$. The black dashed lines are fits to stretched exponential functions, $F_s(q,t) \sim \exp(-(\frac{t}{\tau_\alpha})^\beta)$, where $\tau_\alpha$ is the relaxation time and $\beta$ is the stretching exponent. The dotted lines correspond to the value $F_s(q,t) = \frac{1}{e}$.

A typical snapshot from one of the simulations ($\phi = 0.91$) and the trajectories for a few small- and big-sized cells are displayed in **Appendix 5—figure 2A–D**. The figures reflect the decay in $F_s(q,t)$. Not only are the mobilities heterogeneous, it is also clear that the displacements of the small cells are greater than the large cells.

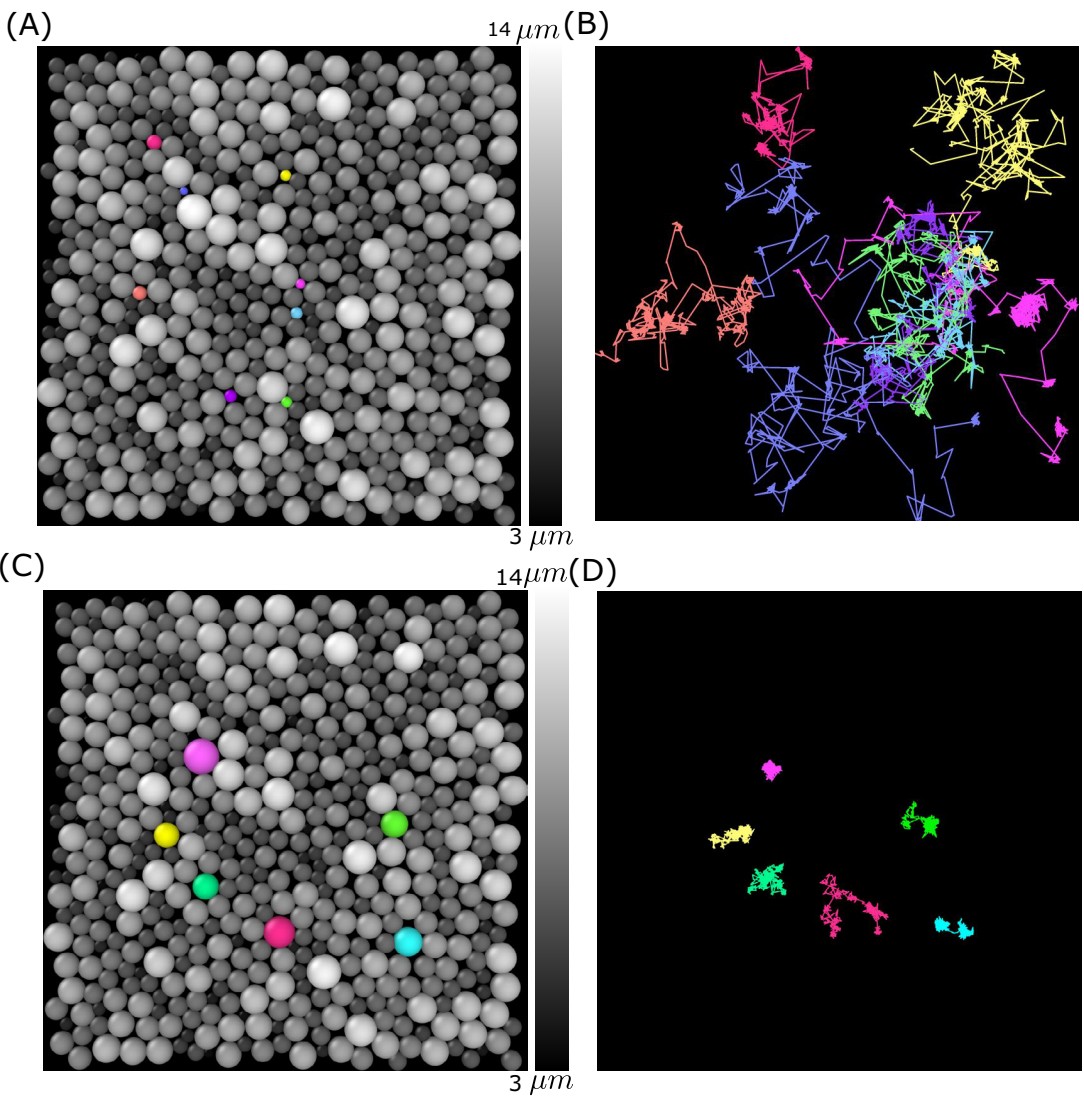

**Appendix 5—figure 2.** Simulation snapshot and trajectories for a few smaller- and bigger-sized cells. (**A**) Cells ($\phi = 0.91$) are colored according to their sizes (gray colors). A few small-sized cells are shown in different colors (pink, blue, orange, purple, cyan, light purple, and yellow). (**B**) The corresponding trajectories are shown over the entire simulation time. (**C**) Similar plot as (**A**) but for a few bigger-sized cells shown in purple, yellow, light green, red, cyan, and green colors. (**D**) Same as (**B**) except the trajectories of the large-sized cells are highlighted. Clearly, the large cells are jammed.

# Appendix 6

## Dynamical changes in local packing fraction cause jammed cells to move

The mobilities of all the cells, even under highly jammed conditions ($\phi \geq \phi_S$), are only possible if the local area fraction changes dynamically. This is a collective effect, which is difficult to quantify because it would involve multi-cell correlation function. As explained in the main text, the movement of a jammed cell can only occur if several neighboring cells move to create space. This picture is not that dissimilar to kinetic facilitation in glass-forming materials (*Hedges et al., 2009*; *Biroli and Garrahan, 2013*). However, in glass-like systems, facilitation is due to thermal excitation, but in the active system, it is self-propulsion that causes the cells to move.

The creation of free space may be visualized by tracking the positions of the nearest-neighbor cells. In *Appendix 6—figure 1*, we display the local free area of a black-colored cell at different times. The top panels show the configurations where the black cell is completely jammed by other cells. The cell, colored in black, can move if the neighboring cells rearrange (caging effect in glass-forming systems) in order to increase the available free space. The bottom panels show that upon rearrangement of the cells surrounding the black cell its mobility increases. Such rearrangement occurs continuously, which qualitatively explains the saturation in viscosity in the multicomponent cell system.

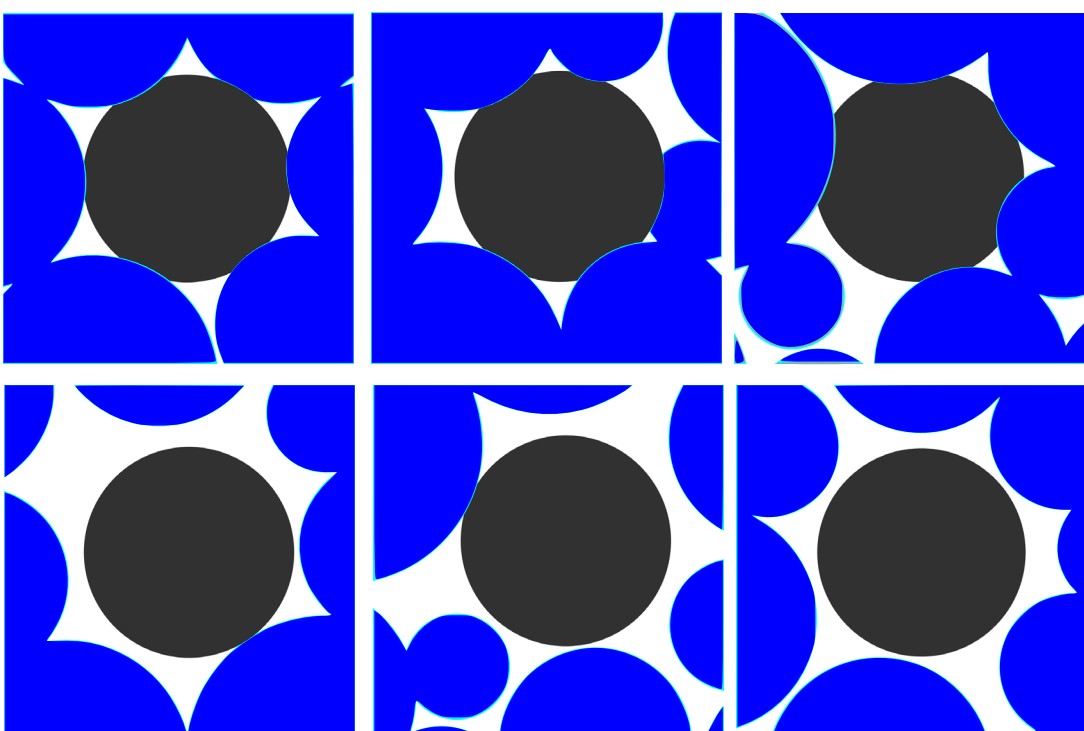

**Appendix 6—figure 1.** Dynamical rearrangement of jammed cells. The changing local environment of a randomly selected cell (black) over time. Top panels: from left to right, $t = 9.41\tau_\alpha$, $10.01\tau_\alpha$, and $25.39\tau_\alpha$. The black-colored cell is completely jammed by other cells. Bottom panels: from left to right, $t = 10.97\tau_\alpha$, $25.44\tau_\alpha$, and $27.49\tau_\alpha$. Dynamical facilitation, resulting in collective rearrangement of the cells surrounding the black cell, enables it to move in the dynamically created free volume.

## Appendix 7

### Finite system size effects

In the main text, we report results for $N = 500$. To asses if the unusual dynamics is not an effect of finite system size, we performed additional simulations with $N = 200$ and $N = 750$. As shown in **Appendix 7—figure 1A and B**, $F_s(q,t)$ saturates at $\phi \geq \phi_S$, which is reflected in the logarithm of $\tau_\alpha$ as a function of $\phi$ (**Appendix 7—figure 1C and D**). The saturation value $\phi_S \sim 0.90$ is independent of the system size. The value of $\phi_0$ ($\approx 0.95$) is also nearly independent of system size. Therefore, the observed dynamics, reflected in the plateau in the viscosity at high $\phi$, is likely not an effect of finite system size.

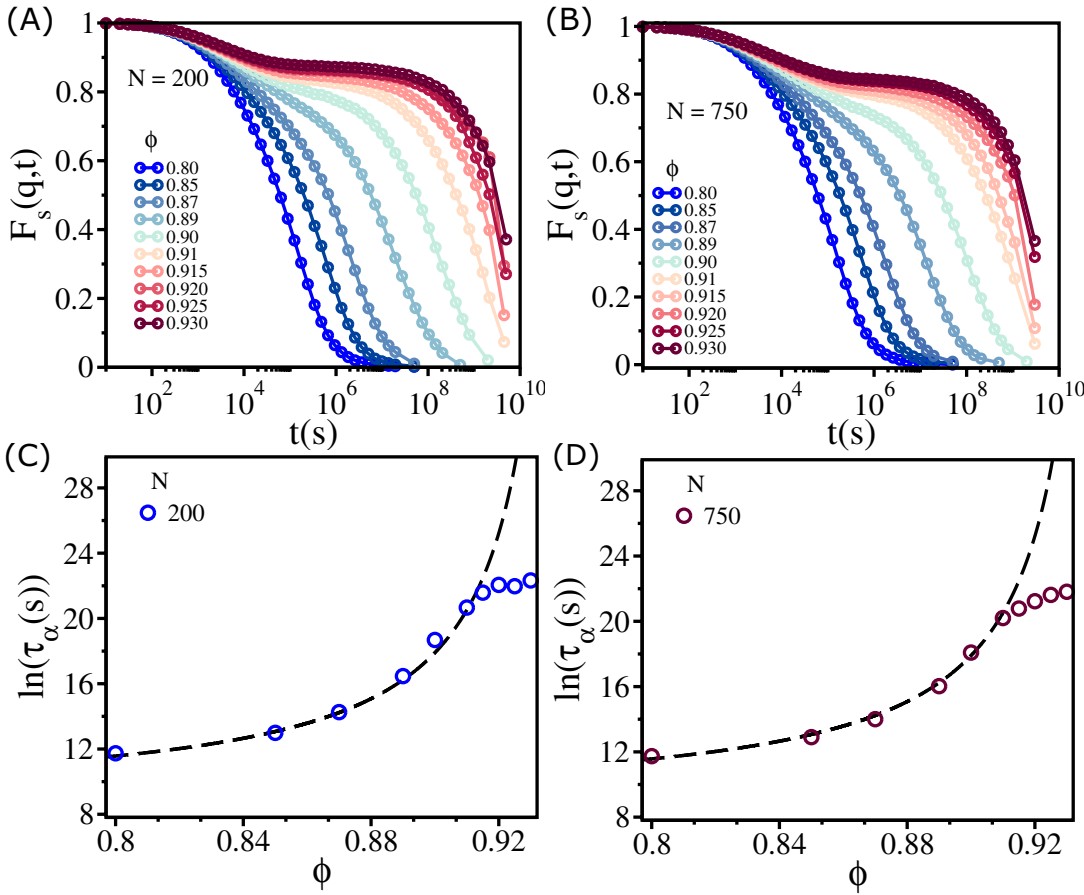

**Appendix 7—figure 1.** Finite size effects. $F_s(q,t)$ for $N = 200$ (**A**) and $N = 750$ (**B**). Logarithm of $\tau_\alpha$ as a function of $\phi$ for $N = 200$ (**C**) and for $N = 750$ (**D**). The dashed lines are the Vogel–Fulcher–Tammann (VFT) fits.

## Appendix 8

### Dependence of viscosity on average coordination number and average connectivity

In the zebrafish blastoderm experiment (*Petridou et al., 2021*), the change in viscosity ($\eta$) was shown as a function of mean connectivity $\langle C \rangle$. To test whether our 2D tissue simulations also exhibit similar behavior, we calculated viscosity as a function of mean coordination number $\langle N_c \rangle$ (defined below), which is equivalent to $\langle C \rangle$. We define the coordination number, $N_c$, as the number of cells that are in contact with a given cell. Two cells with indices $i$ and $j$ are in contact if $h_{ij} = R_i + R_j - r_{ij} > 0$. We calculate $N_c$ for all the cells for each $\phi$ and calculate the histogram $P(N_c)$. The distributions of $P(N_c)$ are well fit by a Gaussian distribution function $A \exp\left[-(\frac{x - \mu}{\sigma})^2\right]$ (*Appendix 8—figure 1A–C*). The calculated mean, $\langle N_c \rangle$, from the fit is linearly related to the cell area fraction $\phi$ (*Appendix 8—figure 1D*). We also calculate the average connectivity $\langle C \rangle$ defined in the following way. Each cell is defined as a node, and an edge is defined as the line connecting two nodes. If a snapshot has $n$ nodes and $m$ edges, then the connectivity is defined as $C = \frac{2m}{n}$ (*Petridou et al., 2021*). We calculate $C$ for all the snapshots for each $\phi$ and estimated its mean value $\langle C \rangle$. We find that $\langle C \rangle$ and $\langle N_c \rangle$ are of similar values (*Appendix 8—figure 2A*), and the dependence of viscosity $\bar{\eta}$ on $\langle N_c \rangle$ and $\langle C \rangle$ is similar to experimental results (*Appendix 8—figure 2B and C*; *Petridou et al., 2021*).

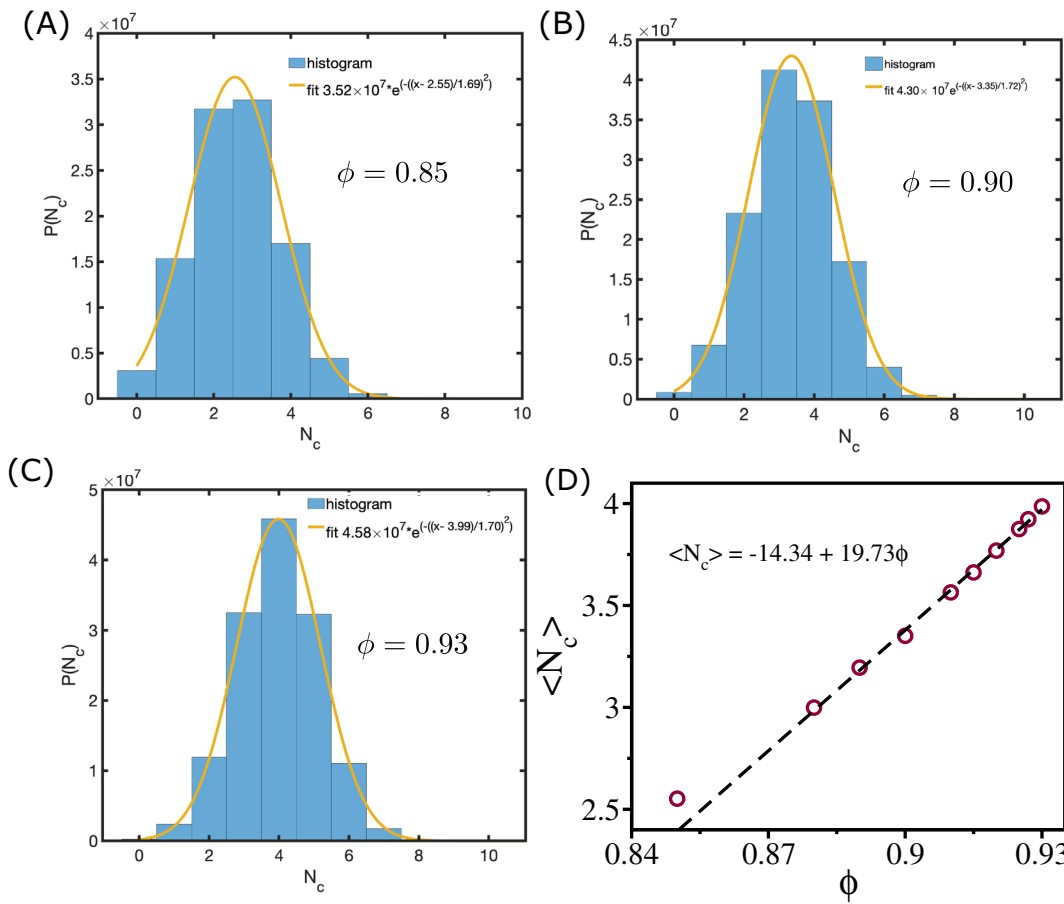

**Appendix 8—figure 1.** Mean coordination number and cell area fraction. (**A—C**) shows the distribution of coordination number $P(N_c)$ for $\phi$ = 0.85, 0.90 and 0.93, respectively. The orange lines are Gaussian fits to the histograms. (**D**) shows mean $\langle N_c \rangle$ as a function of $\phi$. The dashed line shows the linear relationship between them.

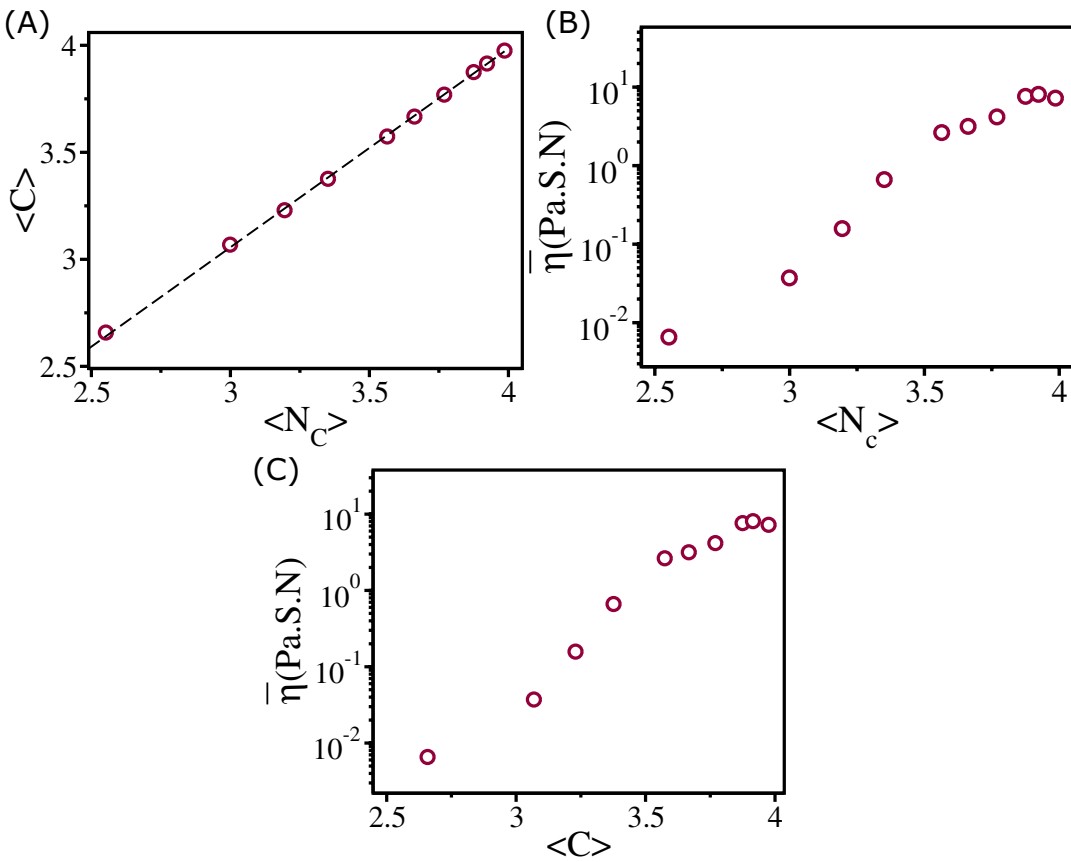

**Appendix 8—figure 2.** Viscosity and coordination number. (**A**) shows $\langle C \rangle$ as a function of $\langle N_c \rangle$. Clearly they are linearly related as shown by the dashed line. Viscosity $\bar{\eta}$ as a function of $\langle N_c \rangle$ (**B**) and $\langle C \rangle$ (**C**).

## Appendix 9

### Connectivity map

To pictorially observe the percolation transition in the simulations, we plot the connectivity map for various values $\phi$ in *Appendix 9—figure 1*. Note that for smaller $\phi \leq 0.89$ the map shows that cells are loosely connected, suggesting a fluid-like behavior. For $\phi \geq 0.89$, the connectivity between the cells spans the entire system, which was noted and analyzed elsewhere thoroughly (*Petridou et al., 2021*). Cells at one side of simulation box are connected to cells at the side. The cell connectivity extends throughout the sample. The transition from a non-percolated state to a percolated state occurs over a very narrow range of $\phi$, which corresponds to the onset of rigidity percolation transition (*Petridou et al., 2021*). It is gratifying that the simple simulations reproduce the experimental observations.

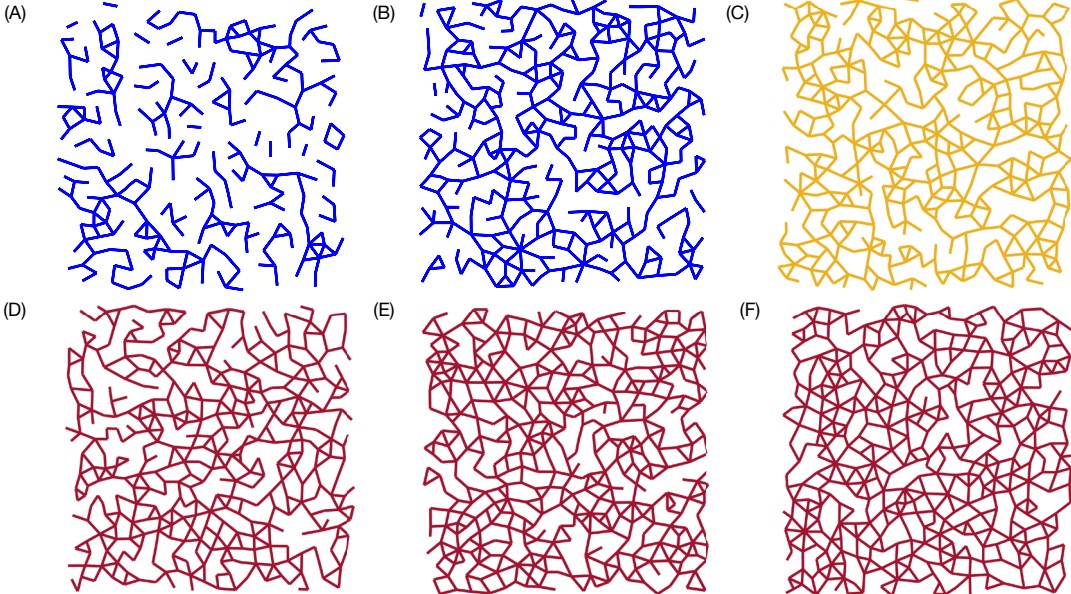

**Appendix 9—figure 1.** Connectivity profile. Connectivity maps for $\phi = 0.80, 0.85, 0.89, 0.90, 0.92$, and $0.93$ are shown in (**A**), (**B**), (**C**), (**D**), (**E**), and (**F**), respectively. For $\phi \geq 0.89$, there is a path that connects the cells in the entire sample. The percolation transition occurs over a very narrow range of $\phi$ (roughly at $\phi \approx 0.89$ ; orange map), which also coincides with the sharp increase in $\eta$, thus linking equilibrium transition to geometric connectivity.

