## [Editor Report · eLife assessment]

This **fundamental** study substantially advances our physical understanding of the sharp increase and saturation of the viscosity of non-confluent tissues with increasing cell density. Through the analysis of a simplified model, this study provides **compelling** evidence that polydispersity in cell size and the softness of cells together can lead to this phenomenon. The work will be of general interest to biologists and biophysicists working on development.

---

## [Referee Report · Joint Public Review]

This paper explores how minimal active matter simulations can model tissue rheology, with applications to the in vivo situation of zebrafish morphogenesis. The authors explore the idea of active noise, particle softness and size heterogeneity cooperating to give rise to surprising features of experimental tissue rheologies (in particular an increase and then a plateau in viscosity with fluid fraction). In general, the paper is interesting from a theoretical standpoint, by providing a bridge between concepts from jamming of particulate systems and experiments in developmental biology. The idea of exploring a free space picture in this context is also interesting. It will be interesting in the future to see whether and how the findings change when considering 3D tissues with less size heterogeneity or how viscosity is impacted by the time scale of measurements.

---

## [Author Response]

The following is the authors’ response to the previous reviews.

We are pleased that Reviewers 1 and 3 have recommended that the revised paper be published.

Reviewer #2For point A: Their preliminary simulation in 3D looks also nice, although it’s referenced in the discussion but not actually included in manuscript - I would advise adding it even under the mention of preliminary.

We appreciate the reviewer for liking our 3D results and suggesting to include them in the manuscript. However, these are preliminary results of our ongoing work. We are yet to establish the corresponding viscosity results quantitatively in the 3D simulations. Because the relationship between viscosity and relaxation time is not (always) linear in glass forming systems, we hesitate to report our results for publication. We hope to report the new results as part of a separate work.

For point B/C: I see some of the points of the authors - although not all of it made it in the main text. I still have some points that puzzle me. For instance, the authors mention that a single value of viscosity (from Green-Kubo) is ”valid for all time scales and amplitude”. This sounds very surprising to me for a complex fluid even at equilibrium: doesn’t it for instance assume linear response (hence small amplitudes)? Fast vs slow probing of a complex medium should also matter (see refs previously mentioned). Related to this, it’s not clear how can self-propulsion not matter if one would shear the system at a finite time scale, given past work on motility-driven unjamming and the mechanism of the authors from facilitation ( wouldn’t shearing at time scales larger vs smaller than the typical time for given cells to spontaneously rearrange from self-propulsion change drastically the effective complex modulus of the system?)

There might be a slight misunderstanding between the reviewer and us when

we say ‘single value of viscosity is valid for all time-scales and amplitude’. Let us explain this point more carefully. In our problem, we are studying the dynamics of a many body system which is undergoing Brownian dynamics where the fluctuation-dissipation theorem need not be valid (as the friction and the selfpropulsion noise strength are not related via Fluctuation-Dissipation Theorem). Now, for us to use the concepts of linear-response (which in the present study are the Green-Kubo relations for the transport coefficients in terms of timecorrelations functions), we need to show that the within the simulation time, the system has reached state that could be described using an “equilibrium” probability measure. This is the precise reason we calculated the ergodicity measure, which is a way to show that all the phase-space have been sampled uniformly under the given Brownian dynamics. This suggests (does not prove) that the system has attained a stationary probability measure (i.e, near equilibrium) for the value of self-propulsion used. Now for this value of self-propulsion, the Green-Kubo relations hold for ‘any time-scale of the simulations’ so that we can perform a time average over the trajectories of the particles (which is an alias of the stationary probability measure under the values of self-propulsion used). If we change the amplitude of the self-propulsion, we need to again compute the ergodicity measure and show the stationarity of the probability measure. If the system is ergodic with respect to the new self-propulsion, we can again use Green-Kubo for the simulations. Note that we will definitely get a different value of viscosity under the new self-propulsion as the shear-stresses generated will be different but the Green-Kubo holds. If the system is not ergodic, for the self-propulsion with the new amplitude, we cannot use Green-Kubo relations. Also a priori, one cannot say what is a large/small amplitude of self-propulsion because it has to be compared with the intrinsic energy scale, which is encoded in the energy function, which is difficult to say without explicit calculations.

This is what we meant when we said, ‘single value of viscosity is valid for all time-scales and amplitude’. It is valid for time-scales of the simulations for a given amplitude of self-propulsion only if the system is ergodic. Note that if the system is not ergodic, then the results of Ref. [14] (in the main text) could be questioned on theoretical grounds, because they were analyzed using 3 the equilibrium rigidity percolation theory. Nevertheless, the authors of Ref. [14] showed that equilibrium phase transition theory works in tissues. For these reasons, we have been, just like the Reviewer, puzzled that equilibrium ideas appear to be valid in the cell system. Additional theoretical work has to be done to clarify these links in tissues. Although this is not the last word, we hope this clarifies our view point.

For point D: I agree with the simplicity argument, although the added sentence from the discussion “Furthermore, the physics of the dynamics in glass forming materials does not change in systems with and without attractive forces” seems a bit strong given works like Lois et al., PRL, 2008 or Koeze et al, PRL, 2018 finding fundamentally different physics of jamming with or without adhesion.In the two cited papers the authors only consider equilibrium transitions in systems with attraction using computer simulations. Apparently, jamming properties depend on the strength of attraction. There are no attempts to characterize the dynamics, the focus of our work.

What we meant is that any universal relations, such as the Vogel-FulcherTammann relation, would still be valid. Of course, non-universal quantities such as glass transition temperature Tg or fragility will change. In our case, changing the adhesion strength would change ϕS, and the parameters in the VFT. However, our contention is that the overall finding that increase in viscosity followed by saturation is unlikely to change. We have added some clarifying statements in the manuscript to make this clear.